

# Automatic Optical Depth Parametrization in Radiative Transfer Model RTTOV v13 via LASSO-Induced Sparsity for Satellite Data Assimilation

Franklin Vargas Jiménez [1,2] and Juan Carlos De los Reyes [1,2]

[1]Research Center in Mathematical Modelling and Optimization (MODEMAT), Ecuador.
[2]Department of Mathematics, Escuela Politécnica Nacional, Ecuador.

**Correspondence:** Juan Carlos De los Reyes (juan.delosreyes@epn.edu.ec)

**Abstract.** The assimilation of satellite spectral sounder data requires fast and accurate radiative transfer models for retrieving surface and atmospheric variables. This study proposes a novel methodology to automatically parameterize atmospheric optical depths within the RTTOV version 13 scheme using statistical thresholds across pressure levels and LASSO regression to induce sparsity. Numerical experiments with VIIRS infrared channels demonstrate that this approach significantly reduces
computational costs while maintaining accuracy. The sparsity also facilitates the automatic selection of absorbing gases and predictors by channel and pressure level, making it particularly effective for multispectral instruments with numerous atmospheric variables. These findings highlight the potential of sparse regression methods to enhance the efficiency of radiative transfer models for satellite data assimilation.

## 1 Introduction

In satellite data assimilation and remote sensing retrieval, as well as their applications in numerical weather prediction (NWP), the radiative transfer equation (RT) is the principal model used to retrieve global atmospheric variables, such as temperature and trace gases concentrations, including water vapor, ozone, carbon dioxide, and other atmospheric constituents. This is achieved by utilizing top of the atmosphere (TOA) radiance measurements from satellite sounders operating across different channels of the electromagnetic spectrum. The numerical implementation of the RT equation as a forward model can primarily be carried
out using two approaches: Line-by-Line Radiative Transfer models and Fast Radiative Transfer models (Fast-RT).

Line-by-line models simulate satellite radiance by rigorously integrating atmospheric physics and chemical phenomena. These models are highly accurate in replicating the precision of modern instruments, such as hyperspectral sounders like AIRS, CrIS and IASI. However, they are characterized by significant computational demands in terms of CPU time and memory, making them impractical for use in data assimilation. Some of the most well-known models in this category include: LBLRTM,
developed at Atmospheric and Environmental Research, Inc. (AER) Clough et al. (1992); Clough and Iacono (1995); Clough et al. (2005); AMSUTRAN, developed at the Met Office (UK) Turner et al. (2019); and GENLN2, developed at the National Center for Atmospheric Research (NCAR) Edwards (1992). A comparison between LBLRTM and GENLN2 is presented in Matricardi (2007). Another software worth mentioning is kCARTA DeSouza-Machado et al. (2020), a pseudo Line-by-Line





model that uses precomputed and compressed physically intensive processes in RT model to compute radiances more quickly
while maintaining accuracy.

On the other hand, the most common Fast-RT models estimate the expected radiance in a channel (what a sensor actually
measures) and are typically based on statistical approaches. In these models, the complex and computationally costly physical
processes of RT model, the calculation of atmospheric transmittances, are parameterized using statistical models and trained
with output from Line-by-Line software on real atmospheric profile databases. The parameters are adjusted using standard
linear regression models or other machine learning techniques. While these methods sacrifice a small degree of accuracy, they
significantly reduce computational costs, making them practical for use in data assimilation. Some of the most well-known
models in this category include: OPTRAN, developed by the NESDIS-NCEP community McMillin et al. (1995a); Kleespies
et al. (2004); McMillin et al. (2006); The Joint Center for Satellite Data Assimilation (JCSDA) Community Radiative Transfer
Model(CRTM) Han et al. (2006); Chen et al. (2008); and the Radiative Transfer for TOVS model (RTTOV), see Saunders
et al. (2018) and the references cited therein. Other studies using statistical approaches include Matricardi (2010), which
incorporates principal component analysis in RTTOV, as well as Krishnan et al. (2012), Cao et al. (2021), Stegmann et al.
(2022), Mauceri et al. (2022), and Su et al. (2023), which apply machine learning techniques for parametrization, feature
reduction, and sampling strategies.

Even though RTTOV is more efficient than line-by-line models, it remains prohibitively expensive for operational use in
small to medium-sized agencies[1]. Indeed, in current Fast RT models based on linear regression, such as OPTRAN and RTTOV,
training is performed separately for each gas type and pressure level, resulting in an over-parametrization of the RT model,
similar to models based on neural networks. To reduce the number of parameters and make the evaluation of the trained RT
model further less computationally expensive, it is essential to carefully select the most significant gases for each spectral
channel of each instrument type, reduce the number of pressure levels, and implement other ad hoc strategies. These decisions
must account for the multitude of possible combinations and trade-offs, which is why large meteorological agencies rely on
expert teams to identify an optimal configuration of parameters and gases for the Fast RT model.

One promising approach to reducing the number of parameters without relying on expert committees is the use of optimization methods that induce sparsity in the parameters. In particular, the use of LASSO regression, a regularization method that
penalizes the regression coefficients with the $\ell_1$-norm, has proven effective for variable selection and model complexity reduction in various large-scale applications. In the context of radiative transfer, LASSO regression was applied by Cardall et al.
(2023) to improve and estimate parameters in water quality monitoring models with optically complex properties. In Li et al.
(2020), the authors proposed an algorithm for detecting hazardous clouds using passive infrared remote sensing technology
with variable selection. Other studies that combine or compare LASSO with machine learning methods for remote sensing
include: the removal of redundant features in PolSAR and optical images Hong and Kong (2021); estimation of aboveground
forest biomass with variable selection Wang et al. (2022a); identification of important environmental variables for retrieving
soil moisture content Wang et al. (2022b); evaluation of the accuracy and generalization capacity of grassland models Smith
et al. (2023); and a comparison of different machine learning methods for predicting soybean yield Joshi et al. (2023).

---

[1]This is the case for Ecuador's METEO operational system, which currently relies on an HPC with only 700 cores.



Building on this approach, in this paper we target the automatic selection of gases and parameters in Fast RT models by inducing sparsity in the parameters using LASSO regression. We propose a parametrization of transmittances based on statistical thresholds to automatically select the appropriate gases by channel and pressure level, and to induce sparsity in the parameters by replacing the classical regression problem with a LASSO problem within the RTTOV framework. The proposed methodology is tested with VIIRS infrared channels, and the results are compared with the standard RTTOV model. To the best of the authors' knowledge, this is the first time that LASSO regression has been applied to the RTTOV model to automate the selection of gases and parameters.

## 1.1 Organization of the Manuscript

This manuscript is organized as follows: Section 2 outlines the theoretical framework for the RT equation in Line-By-Line models. Section 3 details the general scheme of Fast-RT methods, focusing on RTTOV. Section 4 introduces the proposed transmittance parametrization using statistical inference and LASSO regression model. Section 5 presents the experimental settings and numerical results comparing RTTOV with the proposed method. Finally, Section 6 offers conclusions of the performance of the proposed approach.

## 2 Radiative Transfer Equation

The monochromatic radiative transfer equation for the upwelling radiance in a clear sky, without solar radiation contribution, for a non-scattering atmosphere and in local thermodynamic equilibrium, is given by:

$$I(\nu,\theta) = \tau_s(\nu,\theta)\epsilon_s(\nu,\theta)B(\nu,T_s) + \int_{\tau_s}^{1} B(\nu,T(p))\,d\tau + (1-\epsilon_s(\nu,\theta))\tau_s^2(\nu,\theta)\int_{\tau_s}^{1}\frac{B(\nu,T(p))}{\tau^2}\,d\tau, \tag{1}$$

where $I(\nu,\theta)$ represents the monochromatic TOA radiance, at wave number $\nu$ and the satellite zenith angle $\theta$. $B(\nu,T)$ denotes the Planck function, where $T$ is the layer temperature in Kelvin. The layer-to-space atmospheric transmittance is given by $\tau = \tau(\nu,\theta,p,T,q)$, where $p$ is the pressure (hPa) and $q$ is the concentration (ppmv) of the constituent gases of the atmosphere. In this equation, $T_s$ represents the surface skin temperature, $\epsilon_s$ is the surface emissivity, and $\tau_s$ is the surface transmittance.

The first term in Eq. (1) is the upwelling radiance emitted from the surface, the second term is upward atmospheric radiance emission, and the last term is the downward atmospheric radiance emission reflected by the surface, assuming specular reflection. Surface emissivity can be close to 1 for $\nu$ between 714-1250 $cm^{-1}$ and for surfaces such as bodies of water, ice and healthy plant leaves, carbon powder, allowing the last term to be discarded.

The radiance measured by a satellite instrument is polychromatic in nature and can be simulated by convolving the monochromatic radiance from Eq. (1) with the instrument's Normalized Spectral Response Function (NSRF), which accounts for the efficiency within the channel $[\nu_a, \nu_b]$. This is given by:

$$\overline{I}(\nu^*,\theta) = \int_{\nu_a}^{\nu_b} \phi(\nu,\nu^*)I(\nu,\theta)\,d\nu, \tag{2}$$



where $\phi(\nu,\nu^*)$ is the NSRF, representing the sensitivity to radiance within the spectral range $[\nu_a,\nu_b]$, with $\nu^* \in [\nu_a,\nu_b]$ representing the centroid of the response. The NSRF characterizes how the detectors and spectral components integrate the incoming radiance. Its shape is determined by factors such as the spectral bandwidth, the characteristics of any filters employed, and it

can vary over time as the instrument degrades. The function is normalized such that:

$$\int_{\nu_a}^{\nu_b} \phi(\nu,\nu^*)\, d\nu = 1.$$

Using the expression (2) in Eq. (1), the polychromatic radiance for the spectral channel identified with $\nu^*$, assuming $\epsilon_s = 1$, can be written as (see Weinreb et al. (1981)):

$$\overline{I}(\nu^*,\theta) = \overline{\tau}_s(\nu^*,\theta)B(\nu,T_{es}) + \int_{\tau_s}^{1} B(\nu^*,T_e(p))\, d\overline{\tau}, \tag{3}$$

where $T_{es}$ and $T_e$ are the so-called (Superficial) Effective Temperatures, obtained empirically by linear regression using measured temperatures $T$, to correct the fact that (3) does not necessarily hold for $\nu^*$ using directly $T_s$ and $T$, but rather for some unknown $\nu \in [\nu_a,\nu_b]$.

In Eq. (3), $\overline{\tau}$ represents the layer-to-space atmospheric polychromatic transmittance, which is given by:

$$\overline{\tau}(\nu^*,\theta,p,T,q) = \int_{\nu_a}^{\nu_b} \phi(\nu,\nu^*)\tau(\nu,\theta,p,T,q)\, d\nu. \tag{4}$$

## 2.1 Atmospheric Transmittances

The transmittance in a gaseous medium is described by the Beer-Lambert law as follows:

$$\tau = e^{-d},$$

where $d = d(\nu,\theta,p,T,q)$ is the optical depth of the medium.

The transmittance in the monochromatic case results from multiplying the individual transmittances originating from each

atmospheric absorption source. Among the most important sources are: the spectral absorption lines of absorbent gases such as $H2O$, $O3$, $CO2$, $CO$, $N2O$, $CH4$, $SO2$ and other gases; continuum spectral absorption such as water vapor self-broadened and foreign-broadened; collision induced bands; aerosol extinction, among other types of absorbances.

The monochromatic layer-to-space optical depth for the spectral absorption line due to a set of $s$ relevant gases $\mathbf{g}_1, \mathbf{g}_2, \ldots, \mathbf{g}_s$ that most contribute to line absorption, is given by:

$$d(\nu,\theta,p,T,q) = -\frac{sec(\theta)}{g} \sum_{l=1}^{s} \int_{0}^{p} K^{\mathbf{g}_l}(\nu,p',T(p'))q^{\mathbf{g}_l}(p')\, dp', \tag{5}$$

where $g$ is the gravitational acceleration, $K^{\mathbf{g}_l}$ and $q^{\mathbf{g}_l}$ are the absorption function and the concentration of the gas $\mathbf{g}_l$ respectively.



In the absorption function, the Voigt line shape is commonly used, representing a convolution of a Gaussian profile and a Lorentzian profile to model Doppler and pressure broadening, respectively, see Lavrentieva et al. (2011). This convolution is
applied to each absorption line and are weighted and summed according to their line strength to produce the absorption function for each gas. Within a channel, the number of these lines can range from thousands to several hundred thousand, making the computation highly demanding. This is further compounded by the computational cost associated with the complex calculation of numerous nested integrals required to obtain the polychromatic radiance.

## 3   Fast Radiative Transfer Model

The most well-known Fast Radiative Transfer Models begin by discretizing the atmosphere into $L$ layers, characterized by the pressure points:

$$p_0 < p_1 < \cdots < p_L,$$

where $p_0$ is the top of atmosphere pressure and $p_L$ is the surface pressure. The calculation of polychromatic radiance is obtained by numerically approximating Eq. (3) using composite trapezoidal integral formulas. However, to do this, the polychromatic
transmittance for each layer needs to be parameterized with simpler models due to the computational expense of using rigorous physical representations. For this purpose, in Fast-RT methods like OPTRAN and RTTOV, the polychromatic optical depth is parameterized and fitted using linear regression models to approximate Eq. (5), following the ideas of McMillin and Fleming McMillin and Fleming (1976); Fleming and McMillin (1977); McMillin et al. (1979). The polychromatic transmittance is then computed by applying the Beer-Lambert law.

### 3.1   Parametrization of Optical Depths and Transmittance

The parametrization of the polychromatic optical depth, in the OPTRAN and RTTOV schemes, from layer $i$ to the top of the atmosphere, for a single channel and for a gas $\mathbf{g}_l$ (or type of absorption), is as follows:

$$d_i^{\mathbf{g}_l} = d_{i-1}^{\mathbf{g}_l} + \sum_{j=1}^{m_l} w_{ij}^{\mathbf{g}_l} X_{ij}^{\mathbf{g}_l}, \qquad d_0^{\mathbf{g}_l} = 0, \quad i = 1, 2, \ldots, L, \tag{6}$$

where $X_{ij}^{\mathbf{g}_l}$ are predictors that depend on view angle, temperature and gas $\mathbf{g}_l$ concentration. The model parameters are $w_{ij}^{\mathbf{g}_l}$ and
$m_l$ is de number of predictor for gas $\mathbf{g}_l$.

In these parametrizations for spectral line absorption, there is one parametrization for a mixture of fixed gases —those whose spatio-temporal concentration variations do not significantly contribute to changes in radiance— and one parametrization for each variable gas, primarily H2O, with the optional inclusion of O3, CO2, N2O, CO, CH4, and SO2. These sets of fixed and variable gases may change depending on the channel. The parametrization of water vapor absorption can optionally separate
into line absorption and continuum absorption.

The polychromatic transmittance of layer $i$ to the top of the atmosphere of the gas $\mathbf{g}_l$ is parameterized by:

$$\tau_{(i,0)}^{\mathbf{g}_l} = \exp(-d_i^{\mathbf{g}_l}), \tag{7}$$



and the corresponding total polycromatic transmittance (4) is approximated by:

$$\tau_{(i,0)}^{TOT} = \prod_{l=1}^{s} \tau_{(i,0)}^{\mathbf{g}_l}. \tag{8}$$

To fit the parameterized transmittance Eq. (6) and (7), using linear regression approach, a database consisting of $M$ atmospheric vertical profiles is used. Each vertical profile contains, for each pressure level, the measurements of temperature and the concentration of a set of $s$ gases that contribute to radiance absorption. This database, derived from historical vertical profiles of atmospheric variables, is homogenized by pressure levels for the observations of these variables, can be expressed as:

$$(p_i, T_{ij}, q_{ij}^{\mathbf{g}_1}, q_{ij}^{\mathbf{g}_2}, \ldots, q_{ij}^{\mathbf{g}_s}), \quad i = 0, 1, 2, \ldots, L, \ j = 1, 2, \ldots, M.$$

These is complemented with the calculation of the polychromatic transmittances obtained for a set of $N$ view angles $\theta_1, \theta_2, \ldots, \theta_N$ and for each gas $\mathbf{g}_l$, which is expressed as:

$$(\tau_{ijk}^{\mathbf{g}_1}, \tau_{ijk}^{\mathbf{g}_2}, \ldots, \tau_{ijk}^{\mathbf{g}_s}), \quad i = 1, 2, \ldots, L, \ j = 1, 2, \ldots, M, \ k = 1, 2, \ldots, N. \tag{9}$$

These are calculated using numerical integration of Eq. (4), with the monochromatic transmittances obtained from Line-by-Line software.

Since the total polychromatic transmittance is not necessarily the product of individual gases polychromatic transmittances, as it is in the monochromatic case, the polychromatic transmittances data (9) are corrected using different strategies, which are shown below.

     The first one is the calculation of the so-called effective polychromatic transmittance by McMillin et al. (1995b) for OPTRAN and adopted in RTTOV up to v12 Saunders et al. (2017). Let $\mathbf{g}_l$ be an individual gas and a set of gases $G \subseteq$
$\{\mathbf{g}_1, \mathbf{g}_2, \ldots, \mathbf{g}_s\} \setminus \mathbf{g}_l$, the effective polychromatic transmittance of gas $\mathbf{g}_l$ is defined by:

$$\tau_{ijk}^{\mathbf{g}_l} = \frac{\tau_{ijk}^{G+\mathbf{g}_l}}{\tau_{ijk}^{G}},$$

where $\tau_{ijk}^{G}$ is the polychromatic transmittance obtained from Eq. (4) for the set of gases $G$ included simultaneously in the Line-by-Line monochromatic transmittance calculation. The term $\tau_{ijk}^{G+\mathbf{g}_l}$ is similar to the previous one, including gases in $G$ and $\mathbf{g}_l$ in the Line-by-Line transmittance calculation.

The second approach Xiong and McMillin (2005) and McMillin et al. (2006), for OPTRAN v7 and adopted in RTTOV v13 Hocking et al. (2021), consists of calculating the polychromatic transmittances for each individual absorber, and the total polychromatic transmittance Eq. (8) is multiplied by a corrective term $\tau_{(0,i)}^{COR}$,

$$\tau_{(i,0)}^{TOT} = \tau_{(i,0)}^{COR} \prod_{l=1}^{s} \tau_{(i,0)}^{\mathbf{g}_l}, \tag{10}$$

which is parameterized similarly to Eq. (6) and (7) with crossed gases predictors. The transmittance for the training of corrective
term is given by:

$$\tau_{ijk}^{COR} = \frac{\tau_{ijk}^{TOT}}{\hat{\tau}_{ijk}^{TOT}},$$





where $\tau_{ijk}^{TOT}$ is the polychromatic Line-by-Line transmittance including all absorber and $\hat{\tau}_{ijk}^{TOT}$ is the polychromatic transmittance predicted by the model (8).

## 3.2 Linear Regresion Problems for Optical Depths

The linear regression problem for fitting Eq. (6) and (7) for a gas $\mathbf{g}_l$ and atmospheric layer $i$ can be written as the optimization problem:

$$(LS_{\mathbf{g}_l}) \quad \min_{\mathbf{w}_i^{\mathbf{g}_l} \in \mathbb{R}^{m_l}} \frac{1}{2MN} \left\| A_i^{\mathbf{g}_l} \mathbf{w}_i^{\mathbf{g}_l} - Y_i^{\mathbf{g}_l} \right\|_2^2 \tag{11}$$

where $A_i^{\mathbf{g}_l} \in \mathbb{R}^{MN \times m_l}$, $Y_i^{\mathbf{g}_l} \in \mathbb{R}^{MN}$, and $\mathbf{w}_i^{\mathbf{g}_l} \in \mathbb{R}^{m_l}$ is the parameter vector. A column $j$ of $A_i^{\mathbf{g}_l}$ contains the values resulting from the predictor $X_{ij}^{\mathbf{g}_l}$ for different values of angle, temperature, and gas $\mathbf{g}_l$ concentration for layer $i$ and for each profile.

Similarly, the entries of $Y_i^{\mathbf{g}_l}$ contain the polychromatic optical depth values for layer $i$.

When counting the total number of parameters in the optical depth parametrization in RTTOV v13, considering 6 variable gases (H2O, O3, CO2, N2O, CO and CH4, with 14, 12, 13, 12, 13 and 11 predictors, respectively), fixed gas (with 9 predictors), and the correction term (with 26 predictors), across 100 atmospheric layers, the total can reach up to 11,000 parameters per channel. This increases with the inclusion of SO2 as a variable gas and the incorporation of the water vapor continuum

parametrization.

Up to now, in all versions of RTTOV, the number of parameters has been reduced by manually selecting the variable absorbing gases for a given channel, reducing the pressure levels to 54 for most multispectral sounders and to 101 for hyperspectral sounders. Additionally, in RTTOV v13, a threshold based on optical depths has been applied to exclude gases per layer in the correction term, among other techniques based on expert knowledge.

## 190 4 A Sparse Parametrization of Optical Depths

In this section, we present a methodology to significantly reduce the number of parameters used in optical depth parametrization within the RTTOV v13 framework. The methodology involves automatically selecting absorbing gases per channel and pressure level, as well as identifying the most important predictors for each atmospheric layer. This approach induces sparsity in the regression parameters by combining two tools: statistical inference to determine whether a given gas at a particular layer

requires no parametrization, a parametrization with a single predictor, or a more complex parametrization as described in Eq. (6). In the latter case, the classic linear regression problem is replaced with a LASSO regression problem to select predictors and induce sparsity in the parameter vectors.

### 4.1 Parametrization Based on Statistical Inference

The aim here is to preprocess the data of the polychromatic transmittances in a channel to determine which atmospheric layers

require optical depth parametrization and to automatically exclude gases that do not significantly contribute to the radiance absorption in that channel. To achieve this, we will use confidence intervals to estimate the true polychromatic transmittances.





For a gas $\mathbf{g}_l$ or correction term in a fixed layer $i$, we construct a confidence interval for the mean of the polychromatic transmittances of the layer $i$. This is given by:

$$[\overline{\tau}_i^{\mathbf{g}_l} - E_i^{\mathbf{g}_l}, \overline{\tau}_i^{\mathbf{g}_l} + E_i^{\mathbf{g}_l}]$$

where

$$E_i^{\mathbf{g}_l} = Z_{1-\frac{\alpha}{2}} \frac{s_i^{\mathbf{g}_l}}{\sqrt{NM}},$$

$\overline{\tau}_i^{\mathbf{g}_l}$ is the mean polychromatic transmittance for layer $i$, considering $M$ angles and $N$ atmospheric profiles, $s_i^{\mathbf{g}_l}$ is the corresponding standard deviation, and $Z_{1-\frac{\alpha}{2}}$ is the critical value of a distribution for a confidence level of $1-\alpha$. Given that the number of data points in each layer is $NM$, which is usually sufficiently large (in our experiments, for $N=6$ angles and

$M = 83$ profiles, $NM = 498$), the standard normal distribution is used to obtain the critical value. Thus, the absolute error in approximating the true value of the polychromatic transmittance of gas $\mathbf{g}_l$ in layer $i$ with $\overline{\tau}_i^{\mathbf{g}_l}$ is at most $E_i^{\mathbf{g}_l}$, with a probability of $\alpha$ that the absolute error exceeds this value. In our case, the confidence level is set to $\alpha = 10^{-6}$.

Based on the above, the following statistical thresholds for optical depth parametrizations are proposed. Let $\epsilon_1$ and $\epsilon_2$ be positive and sufficiently small values, these will be used as thresholds to determine whether $\overline{\tau}_i^{\mathbf{g}_l}$ is close to the true value or

close to 1. Define the mean optical depth for layer $i$ as $\overline{d}_i^{\mathbf{g}_l} = -\ln(\overline{\tau}_i^{\mathbf{g}_l})$, and consider the following three cases:

– *Case I*: If $E_i^{\mathbf{g}_l} > \epsilon_1$, the polychromatic transmittance due to gas $\mathbf{g}_l$ in layer $i$ has high variability with respect to the value of the atmospheric variables in that layer. In this case, the optical depth parametrization follows as in Eq. (6) for layer $i$.

– *Case II*: If $E_i^{\mathbf{g}_l} \leq \epsilon_1$ and $\overline{d}_i^{\mathbf{g}_l} > \epsilon_2$, unlike the previous case, the polychromatic transmittance due to gas $\mathbf{g}_l$ in layer $i$ has low variability with respect to the value of the atmospheric variables in that layer, and can be estimated by $\overline{\tau}_i^{\mathbf{g}_l}$, but is not

close to 1. Thus, the optical depth can be parameterized with a single predictor as follows:

$$d_i^{\mathbf{g}_l} = d_{i-1}^{\mathbf{g}_l} + w_{i0}^{\mathbf{g}_l} X_{i0}^{\mathbf{g}_l},$$

where, $X_{0i} = 1$ and $w_{i0}^{\mathbf{g}_l} = \overline{d}_i^{\mathbf{g}_l}$. If this occurs in all layers, and since the parametrization does not depend on atmospheric variables, the gas $\mathbf{g}_l$ can be included with fixed gases.

– *Case III*: If $E_i^{\mathbf{g}_l} \leq \epsilon_1$ and $\overline{d}_i^{\mathbf{g}_l} \leq \epsilon_2$, the polychromatic transmittance in layer $i$ can not only be estimated by $\overline{\tau}_i^{\mathbf{g}_l}$ but is also

close to 1, meaning that gas $\mathbf{g}_l$ does not cause significant absorbance in this layer. The relative error of approximating $\overline{\tau}_i^{\mathbf{g}_l}$ with 1 is given by:

$$\frac{1 - \overline{\tau}_i}{\overline{\tau}_i} = e^{\overline{d}_i^{\mathbf{g}_l}} - 1 = \overline{d}_i^{\mathbf{g}_l} e^{\xi} \leq \epsilon_2 e^{\epsilon_2} \approx \epsilon_2,$$

for some $\xi \in (0, \overline{d}_i^{\mathbf{g}_l})$. If this condition is met for all layers, then gas $\mathbf{g}_l$ is automatically discarded.





To summarize the above, the parametrization of optical depths based on statistical thresholds is as follows:

$d_0^{\mathbf{g}_l} = 0$

$$d_i^{\mathbf{g}_l} = d_{i-1}^{\mathbf{g}_l} + \begin{cases} \sum_{j=1}^{m_l} w_{ij}^{\mathbf{g}_l} X_{ij}^{\mathbf{g}_l}, & E_i^{\mathbf{g}_l} > \epsilon_1, \\ \overline{d}_i^{\mathbf{g}_l}, & E_i^{\mathbf{g}_l} \le \epsilon_1 \text{ and } \overline{d}_i^{\mathbf{g}_l} > \epsilon_2, \\ 0, & \text{otherwise,} \end{cases} \tag{12}$$

for $i = 1, 2, \ldots, L$. Experimentally, $\epsilon_1 = \epsilon_2 = 10^{-6}$ are used. The transmittances from layer $i$ to the top of the atmosphere are still calculated using (7).

### 4.2 Linear Regression with LASSO

After discarding parameter groups using the previous statistical approach with Eq. (12), in *Case I*, the parameters must be obtained by solving the least squares problem. Alternatively, it is proposed to induce sparsity in the parameter vector $\mathbf{w}_i^{\mathbf{g}_l}$ to discard predictors per atmospheric layer by solving the LASSO problem:

$$(LASSO_{\mathbf{g}_l}) \quad \min_{\mathbf{w}_i^{\mathbf{g}_l} \in \mathbb{R}^{m_l}} \frac{1}{2MN} \|A_i^{\mathbf{g}_l} \mathbf{w}_i^{\mathbf{g}_l} - Y_i^{\mathbf{g}_l}\|_2^2 + \lambda_i \|\mathbf{w}_i^{\mathbf{g}_l}\|_1 \tag{13}$$

where $\lambda_i \ge 0$ is the regularization parameter.

As $\lambda_i \to +\infty$, high sparsity is induced, and as $\lambda_i \to 0$, sparsity is low. Specifically, if $\lambda_i = 0$, the problem reduces to the least squares problem (11). The selection of this regularization parameter is carried out to ensure that there is no significant loss of precision in approximating the transmittance in layer $i$ compared to the least squares problem (11), while achieving a model with fewer parameters. Since there is no prior information to estimate this parameter, a Grid Search strategy is employed.

We start by searching through a sequence $\lambda_i \in \{10^{-3}, 10^{-4}, \ldots, 10^{-12}, 0\}$ in this order, until the following condition is met:

$\text{mse}(\lambda_i) \le 2\text{mse}(0)$,

where

$$\text{mse}(\lambda_i) = \frac{1}{NM} \|A_i^{\mathbf{g}_l} \mathbf{w}_i^{*\mathbf{g}_l}(\lambda_i) - Y_i^{\mathbf{g}_l}\|_2^2$$

and $\mathbf{w}_i^{*\mathbf{g}_l}(\lambda_i)$ is the optimal solution of the LASSO problem (13) with regularization parameter $\lambda_i$.

## 5   Numerical Results

This section aims to evaluate the performance of the proposed parametrization compared to the standard form of RTTOV v13. Specifically, the goal is to study the level of sparsity achieved with the proposal and its impact on accuracy when compared to RTTOV v13 and Line-by-Line calculations using LBLRTM. To accomplish this, the performance of each parametrization is evaluated by measuring the root mean square error (RMSE) of the transmittances compared to the Line-by-Line transmittances, and by assessing the approximation error of the brightness temperature from the Fast-RT against those obtained Line-by-Line.



## 5.1 Experiment settings

For training the RTTOV parametrizations and the proposed sparse variants, six variable gases are considered: $H_2O$, $O_3$, $CO_2$, $N_2O$, $CO$, and $CH_4$. The Fast-RT model can additionally consider $SO_2$ as a variable gas, but here it will be treated as a fixed gas among the total of 22 fixed gases considered. No distinction is made between water vapor absorption lines and continuum absorption. For the viewing angle, we consider 6 path secant angles from 1 to 2.25 with step 0.25 (from $0°$ to $63.61°$).

### 5.1.1 Spectral Response Functions of VIIRS M-bands:

The Visible Infrared Imaging Radiometer Suite (VIIRS) is an instrument on NOAA's Suomi NPP and NOAA-20 satellites, part of the Joint Polar Satellite System (JPSS). It features 16 moderate resolution bands (M-bands) that cover visible and infrared spectra. This study focuses on spectral response functions for bands M7 to M16, which cover the near (NIR), medium (MIR), and long (LIR) infrared ranges. In this study, we use the VIIRS SRF J2 and can be downloaded from the following link: https://ncc.nesdis.noaa.gov/NOAA-21/index.php. Details on the centers and spectral ranges of these bands can be found in Tables 1 and 2 in Cao et al. (2017).

For each channel, the wavenumber $\nu$ and the corresponding SRF values are tabulated. The wavenumber tabulation typically covers a broader spectral range, denoted as $[\nu_a, \nu_b]$, with noisy SRF values at the extremes of this interval. Therefore, the SRF must be truncated to a smaller interval that retains most of the relevant SRF information. Instead of using Tables 1 and 2 from Cao et al. (2017) for our calculations, we utilize channels with a spectral range broader than those. These channels are defined as $[\nu^* - \nu_l, \nu^* + \nu_u]$, where $\nu^*$ is the centroid of SRF in $[\nu_a, \nu_b]$, $\nu_l$ and $\nu_u$ are the tabulated wavenumber values closest to $\nu^*$ below and above, respectively, such that the relative truncation error does not exceed $\epsilon = 9 \times 10^{-4}$. Specifically:

$$(1 - \epsilon) \int_{\nu_a}^{\nu_b} \phi(\nu^*, \nu)\, d\nu \leq \int_{\nu^* - \nu_l}^{\nu^* + \nu_u} \phi(\nu^*, \nu)\, d\nu.$$

The integrals are calculated using the composite trapezoidal rule. The SRF data are then truncated and normalized within this new interval, and the centroid $\nu^*$ is recalculated. The updated channels and centroids are presented in Table 1.



| Band | Centroid ($cm^{-1}$) | Spectral Range ($cm^{-1}$) | IR |
|------|------|------|------|
| M7 | 11525.42 | 11070.96 – 12048.02 | NIR |
| M8 | 8056.98 | 7924.69 – 8170.62 | NIR |
| M9 | 7235.57 | 7134.59 – 7373.52 | NIR |
| M10 | 6199.43 | 5853.32 – 6522.30 | NIR |
| M11 | 4442.00 | 4342.01 – 4549.99 | NIR |
| M12 | 2711.61 | 2545.18 – 2867.98 | MIR |
| M13 | 2489.30 | 2354.64 – 2607.44 | MIR |
| M14 | 1166.76 | 1111.73 – 1235.32 | LIR |
| M15 | 939.82 | 875.89 – 1008.36 | LIR |
| M16 | 839.10 | 782.35 – 896.29 | LIR |

**Table 1.** VIIRS IR M-bands (wavenumber)

By truncating the noisy tails of the SRF in this way, the resulting NSRF for each channel is interpolated using natural cubic splines to be used for calculating polychromatic transmittances with a much finer spectral resolution than the tabulated NSRF data. It can be shown that the error made by approximating the polychromatic transmittance with the truncated NSRF does not exceed $\epsilon$.

### 5.1.2 Vertical profile database ECMWF83:

For training the optical depth parametrization, we use the ECMWF83 database, which includes 83 vertical profiles with temperature and gas concentrations for $H_2O$, $O_3$, $CO_2$, $N_2O$, $CO$ and $CH_4$, across 101 pressure levels, originally created to train RTTOV Matricardi (2008). A separate database with 22 vertical profiles covers fixed gases. These datasets are available from NWP SAF of EUMETSAT and can be downloaded at https://nwp-saf.eumetsat.int/site/.

### 5.1.3 Line-by-Line Transmitances with LBLRTM:

In this study, LBLRTM v12.15.1 (February 2023) will be employed for Line-by-Line calculations. The software uses AER Continuum MT CKD v4.1.1. for continuum models of water vapor and other gases and the AER Line Parameter Database v3.8.1. for line parameters, which consolidates various line spectral databases, primarily HITRAN 2016 Gordon et al. (2017).

The principal parameter in the LBLRTM calculation, to generate the optical depths for training and top-of-atmosphere radiances, are the following:

– The continuum absorption is not activated for isolated gases and fixed gases; it is only activated when all gases are included: the 22 fixed gases plus the 6 variable gases,

– The Voigt profile is chosen for the shape of spectral lines,





- The spectral resolution is set to $d\nu = \bar{\alpha}_\nu/1.5$ where $\bar{\alpha}_\nu$ is the average value of the Voigt halfwidth for the layer. Consequently, the spectral resolution is not homogeneous across channels, achieving an average spectral resolution from $7.1 \times 10^{-3}$ for M7 to $4.1 \times 10^{-4}$ for M16.

- The calculation of optical depths with the software is performed only for the observation point at nadir. For other angles, variations are made directly in the calculation of polychromatic transmittances.

### 5.1.4 RTTOV v13 Settings:

We implemented the transmittance parametrization of RTTOV v13 as described in Saunders et al. (2020), using the same predictors, except for the method of selecting gases per channel, which is detailed below.

In RTTOV v13 in the standard form, regression parameters are obtained by including only the gases that exhibit absorption lines in each channel, as shown in Table 2. In the proposed RTTOV variants, using statistical inference and LASSO regression, all gases are included in the training.

Additionally, there are other criteria for selecting predictors in the correction term and training data by level, which are listed below:

- **Threshold for gases correction term**: Predictors for fixed gases are always included in the correction term. For other gases, predictors for a specific gas in a layer are included only if any of the corresponding optical depths in the training profile for that layer exceed a threshold 0.01 for CH4 and 0.005 for the other gases. As a result, for all the VIIRS channels studied, only predictors for fixed gases and water vapor are included in the correction term.

- **Threshold for Optical Depth Data Training**: Optical depth data in a layer for a gas is omitted if the corresponding transmittance from the layer to the surface is less than $3 \times 10^{-6}$. As a result, only channel M10 is affected by this selection criterion.

| Channels | Gases |
|---|---|
| M7 | H2O, CO2, CH4 |
| M8 | H2O, CO2, CO, CH4 |
| M9 | H2O, CO2, NO2, CH4 |
| M10-M11 | H2O, O3, CO2, N2O, CO, CH4 |
| M12-M16 | H2O, O3, CO2, N2O, CH4 |

**Table 2.** Gases considered in RTTOV v13 for VIIRS M-bands.

### 5.2 Sparsity Pattern in the parametrization of optical depths

Table 3 summarizes the percentage of non-zero parameters (%NZ) based on a total of 11,000 parameters (worse case) for each type of optical depth model: RTTOV v13 in its standard form (RTTOV13), RTTOV v13 with statistical thresholds and standard regression (RTTOV13+SI), and RTTOV v13 with statistical thresholds and LASSO regression (RTTOV13+SI+LASSO). Tables 4, 5, and 6 provide details on the number of non-zero parameters (NNZ) used for each gas type and correction factor.





In Table 3, the increase in sparsity for the proposed parametrizations compared to the general RTTOV v13 scheme is evi-
dent. RTTOV v13 induces sparsity by manually selecting gases and using criteria based on optical depth thresholds to include
predictors in the correction factor. Comparing the different approaches, in the best-case scenario for channel M7, the spar-
sity level of RTTOV v13 (65.45%) improves to 94.20% with RTTOV13+SI+LASSO. Conversely, in the worst-case scenario
for channels M10 and M11, the sparsity level of RTTOV v13 (20%) increases to 86.20% and 90.02% respectively, with RT-
TOV13+SI+LASSO. This suggests that the computational cost of evaluating parameterized transmittances is significantly and
proportionally reduced with the proposed parametrization.

| Fast-RT | M7 | M8 | M9 | M10 | M11 | M12 | M13 | M14 | M15 | M16 |
|---|---|---|---|---|---|---|---|---|---|---|
| RTTOV13 | 34.55 | 46.36 | 59.77 | 80.00 | 80.00 | 69.95 | 68.18 | 70.00 | 69.41 | 69.77 |
| SI | 8.80 | 10.05 | 24.64 | 24.79 | 25.13 | 32.98 | 31.06 | 37.76 | 34.26 | 27.30 |
| LASSO | 5.80 | 6.50 | 12.08 | 13.80 | 9.98 | 16.78 | 14.02 | 16.32 | 20.72 | 15.43 |

Table 3. Percentage of nonzero parameters in RTTOV v13 for each channel, for the standard, RTTOV v13 + SI, and RTTOV v13 + SI +
LASSO parameterizations.

In Tables 5 and 6, the effectiveness of introducing statistical thresholds to discard irrelevant gases by channel is clear com-
pared to Table 4. The number of non-zero parameters below 100 for a specific gas corresponds to case II of the statistical
threshold parametrization, suggesting that the corresponding gas can be included with the fixed gases.

| Gas | M7 | M8 | M9 | M10 | M11 | M12 | M13 | M14 | M15 | M16 |
|---|---|---|---|---|---|---|---|---|---|---|
| FIX | 900 | 900 | 900 | 900 | 900 | 900 | 900 | 900 | 900 | 900 |
| H2O | 1400 | 1400 | 1400 | 1400 | 1400 | 1400 | 1400 | 1400 | 1400 | 1400 |
| O3 | 0 | 0 | 0 | 1200 | 1200 | 1200 | 1200 | 1200 | 1200 | 1200 |
| CO2 | 0 | 0 | 1300 | 1300 | 1300 | 1300 | 1300 | 1300 | 1300 | 1300 |
| N2O | 0 | 0 | 1200 | 1200 | 1200 | 1200 | 1200 | 1200 | 1200 | 1200 |
| CO | 0 | 1300 | 0 | 1300 | 1300 | 0 | 0 | 0 | 0 | 0 |
| CH4 | 1100 | 1100 | 1100 | 1100 | 1100 | 1100 | 1100 | 1100 | 1100 | 1100 |
| COR | 400 | 400 | 675 | 400 | 400 | 595 | 400 | 600 | 535 | 575 |

Table 4. Number of nonzero parameters by gas type and channel in RTTOV v13.

| Gas | M7 | M8 | M9 | M10 | M11 | M12 | M13 | M14 | M15 | M16 |
|---|---|---|---|---|---|---|---|---|---|---|
| FIX | 18 | 247 | 0 | 0 | 15 | 145 | 52 | 134 | 494 | 615 |
| H2O | 619 | 618 | 1374 | 618 | 604 | 775 | 576 | 802 | 687 | 716 |
| O3 | 0 | 0 | 0 | 0 | 0 | 655 | 19 | 1120 | 1096 | 488 |
| CO2 | 0 | 56 | 0 | 1142 | 0 | 212 | 724 | 0 | 1213 | 911 |
| N2O | 0 | 0 | 0 | 0 | 897 | 640 | 1024 | 1024 | 33 | 0 |
| CO | 0 | 0 | 0 | 0 | 0 | 0 | 0 | 0 | 0 | 0 |
| CH4 | 47 | 0 | 778 | 768 | 995 | 964 | 819 | 678 | 0 | 0 |
| COR | 362 | 319 | 596 | 267 | 296 | 463 | 350 | 545 | 340 | 409 |

Table 5. Number of nonzero parameters by gas type and channel in RTTOV13+SI.



| Gas | M7 | M8 | M9 | M10 | M11 | M12 | M13 | M14 | M15 | M16 |
|-----|----|----|----|-----|-----|-----|-----|-----|-----|-----|
| FIX | 18 | 220 | 0 | 0 | 15 | 103 | 52 | 103 | 386 | 405 |
| H2O | 411 | 393 | 743 | 329 | 396 | 461 | 346 | 473 | 419 | 357 |
| O3 | 0 | 0 | 0 | 0 | 0 | 387 | 19 | 680 | 596 | 277 |
| CO2 | 0 | 56 | 0 | 651 | 0 | 212 | 580 | 0 | 748 | 603 |
| N2O | 0 | 0 | 0 | 0 | 109 | 100 | 139 | 139 | 33 | 0 |
| CO | 0 | 0 | 0 | 0 | 0 | 0 | 0 | 0 | 0 | 0 |
| CH4 | 47 | 0 | 318 | 478 | 527 | 605 | 381 | 307 | 0 | 0 |
| COR | 240 | 181 | 306 | 128 | 94 | 172 | 172 | 241 | 204 | 194 |

**Table 6.** Number of nonzero parameters by gas type and channel in RTTOV13+SI+LASSO.

It is observed that in channels M7-M9, the use of parameters in RTTOV13 is lower compared to channels M10-M16, which exhibit a higher level of parameter usage. In channels M10-M16, RTTOV13 has an average parameter usage of 72.47% (M10-M16 %NNZ average), justifying the need to discard gases and predictors to reduce the computational impact of evaluating transmittance once the model is trained. This is achieved by incorporating statistical thresholds that automatically discard gases by channel or across all channels (as with CO) and by pressure level where the gas concentration is not relevant, reducing the average parameter usage to 30.38%. Additionally, replacing classical linear regression with LASSO regression, further reduces parameter usage to 15.29% by discarding predictors used by gas and by each level.

To illustrate in more detail, we reference channels M11 and M12 and compare the sparsity patterns among the three parametrizations in Fig. 1 and 2. For RTTOV13+SI+LASSO and the remaining channels, see Appendix Figs. A1 and A2. The numbering of predictors and correctors follows RTTOV v13 Saunders et al. (2020), except for predictor 0, which corresponds to the predictor in Case II of the statistical inference proposal. Each column represents the parameters of a predictor for each pressure level, and each point in a column represents a non-zero parameter associated with that predictor at the corresponding pressure level.

In the middle Fig. 1, note that gases O3, CO2, and CO are discarded and FIX gas only needs one predictor. Meanwhile, gases H2O, N2O, and CH4 exhibit block-like sparsity patterns from surface pressure approximately 200 hPa, 19 hPa, and 0.8 hPa, where concentrations of these gases are important and cause significant radiance absorption. Fixed gases also show block-like sparsity patterns in the correction term. For these gases with block-like sparsity patterns, replacing classical linear regression with LASSO regression (bottom figure) clearly discards some predictors across all levels or shows them as less relevant, as seen in the sparsity patterns for N2O, CH4, and fixed gas correctors. However, H2O still shows sparsity, but it is difficult for this channel to determine if any predictor can be discarded at all levels due to the importance of this gas and the strong non-linear relationship among the secant angle, temperature, and gas concentration in the predictors defined for it.







**Figure 1.** Sparsity pattern for channel M11, comparing RTTOV v13 (top), RTTOV v13 + SI (middle), and RTTOV v13 + SI + LASSO (bottom).

For channel M12, shown in Fig. 2, only one gas, CO, is automatically discarded, which was already known a priori due to the fact that this gas does not have absorption lines in this channel. However, with the proposed statistical threshold parametrization, the block sparsity structure of the predictors and correctors for each gas at different atmospheric pressure levels, where they are relevant for absorption, is still evident (see the middle figure). In this channel, $CO_2$ as a variable gas is relevant at high pressure levels, approximately above 767 hPa. Regarding the use of LASSO regression for these pressure levels, where the





atmospheric variables of each gas are important, it continues to show that some predictors can be entirely discarded or are less relevant, as observed for gases O3, N2O, CH4, and the correctors for fixed gases and H2O. For H2O, some predictors begin to lose relevance in this channel, showing a sparser structure by column compared to what was observed in channel M11.

**Figure 2.** Sparsity pattern for channel M12, comparing RTTOV v13 (top), RTTOV v13 + SI (middle), and RTTOV v13 + SI + LASSO (bottom).

A similar analysis can be conducted for each channel, as referenced in the appendix, where Fig. A1 and A2 display the sparsity patterns for all channels using RTTOV13+SI+LASSO. From these figures, it can be appreciated which gases are





relevant for each channel, the pressure level ranges where they are important, and which predictors are most relevant for reconstructing the transmittance for each gas.

### 5.3  Validation of transmittances

To validate the proposed RTTOV v13 variants, we calculated the root mean square error (RMSE) of the total transmittance for all atmospheric layers, vertical profiles, and viewing angles, as shown in the following formula:

$$\text{RMSE} = \left( \frac{1}{LMN} \sum_{i=1}^{L} \sum_{j=1}^{M} \sum_{k=1}^{N} (\tau_{ijk}^{TOT} - \tilde{\tau}_{ijk}^{TOT})^2 \right)^{\frac{1}{2}},$$

where $L = 100$, $M = 83$, and $N = 6$. Here, $\tau_{ijk}^{TOT}$ and $\tilde{\tau}_{ijk}^{TOT}$ represent the polychromatic transmittances calculated using LBLRTM optical depths and their corresponding approximations obtained from Eq. (10) using the training data. The results are shown in Table 7.

| Fast-RT | M7 | M8 | M9 | M10 | M11 | M12 | M13 | M14 | M15 | M16 |
|---------|-----|-----|-----|------|------|------|------|------|------|------|
| RTTOV13 | 0.043 | 0.071 | 5.919 | 0.062 | 1.648 | 0.223 | 1.880 | 1.826 | 3.663 | 4.574 |
| SI | 0.044 | 0.072 | 5.920 | 0.067 | 1.649 | 0.223 | 1.881 | 1.831 | 3.662 | 4.574 |
| LASSO | 0.063 | 0.109 | 5.535 | 0.078 | 2.046 | 0.298 | 1.966 | 1.643 | 3.364 | 3.597 |

**Table 7.** RMSE of total transmittance for each channel, scaled by $10^{-3}$, for the proposed RTTOV v13 variants.

In Table 7, the RMSE for transmittance generally errors ranges between $O(10^{-5})$ and $O(10^{-3})$ across all channels and Fast-
RT methods. Comparing the error of the three methods by channel, the order of magnitude remains the same, except for the RTTOV13+SI+LASSO method for channel M8, which shows an increase in error by one order of magnitude. When comparing RTTOV13 with RTTOV13+SI, the difference in errors ranges between $O(10^{-7})$ and $O(10^{-6})$ for all channels. Comparing RTTOV13 with RTTOV13+SI+LASSO, the error difference increases by between $O(10^{-5})$ and $O(10^{-4})$ for channels M7, M8, and M10-M12, while it decreases by between $O(10^{-6})$ and $O(10^{-4})$ for channels M9 and M13-M16. This suggests that
the inclusion of statistical thresholds and LASSO regression in RTTOV v13 slightly affects the accuracy of the transmittance approximation, either improving or worsening it, but the overall variation in error remains negligible.

### 5.4  Validation of brightness temperatures

To achieve a higher level of validation for the proposed transmittance parametrization, the brightness temperatures (BT) of the profiles used for training are calculated. The approximated brightness temperatures at the top of the atmosphere were
calculated using polychromatic radiances from Eq. (3), applying the approximate transmittances provided by the RTTOV v13 scheme and the proposed variants, separately. To compare these results, brightness temperatures at the top of the atmosphere were calculated using the polychromatic radiances with Eq. (2), using the monochromatic radiances calculated with LBLRTM. In all cases, the integrals were approximated using composite trapezoidal formulas, with the spacing determined by the pressure levels of the data. In each case, the resulting brightness temperatures were averaged over all profiles and viewing angles. The



relative errors in BT obtained with the Fast-RT models and those obtained with LBLRTM were then calculated, which are shown in Table 8. The maximum relative error for brightness temperature, determined for each profile and viewing angle, is presented in Table 9.

| Fast-RT | M7 | M8 | M9 | M10 | M11 | M12 | M13 | M14 | M15 | M16 |
|---------|-----|-----|------|------|------|------|------|------|-------|------|
| RTTOV13 | 3.708 | 0.411 | 9.459 | 2.202 | 3.204 | 0.585 | 5.990 | 5.384 | 11.254 | 8.592 |
| IS | 4.977 | 0.619 | 9.477 | 2.281 | 3.188 | 0.564 | 6.001 | 5.421 | 11.248 | 8.585 |
| LASSO | 4.971 | 0.669 | 9.469 | 2.304 | 3.782 | 0.614 | 6.220 | 6.681 | 11.578 | 8.970 |

**Table 8.** Average Relative Errors in Brightness Temperature (K), scaled by $10^{-4}$, between the Fast-RT and LBLRTM models.

| Fast-RT | M7 | M8 | M9 | M10 | M11 | M12 | M13 | M14 | M15 | M16 |
|---------|-----|-----|--------|------|------|------|------|------|------|------|
| RTTOV13 | 2.486 | 0.649 | 12.637 | 0.896 | 1.915 | 0.401 | 3.492 | 2.719 | 6.940 | 4.896 |
| IS | 9.652 | 2.273 | 12.637 | 1.401 | 1.924 | 0.360 | 3.502 | 2.713 | 6.959 | 4.877 |
| LASSO | 9.749 | 2.270 | 3.865 | 1.401 | 3.080 | 0.487 | 2.901 | 3.256 | 8.279 | 5.811 |

**Table 9.** Maximun Relative Errors in Brightness Temperature (K), scaled by $10^{-3}$, between the Fast-RT and LBLRTM models.

In Table 8, a similar behavior is observed in the errors when approximating transmittances. The average relative error of brightness temperature generally ranges from $O(10^{-5})$ to $O(10^{-3})$ across all channels and Fast-RT methods. The order of magnitude of the average relative error remains consistent when comparing the three methods by channel. The differences in average relative BT errors between RTTOV13 and RTTOV13+SI range from $O(10^{-7})$ to $O(10^{-4})$, while those between RTTOV13 and RTTOV13+SI+LASSO range from $O(10^{-6})$ to $O(10^{-4})$ across all channels. Table 9 shows that the maximum relative errors of BT range from $O(10^{-4})$ to $O(10^{-2})$ across all channels and Fast-RT methods. When comparing the maximum absolute error by channel for the three methods, the errors remain of the same order of magnitude for channels M7 and M11-M16, for channels M8 and M10, the standard RTTOV13 version has an order of magnitude lower error compared to the proposed variants, and in channel M9, RTTOV13+IS+LASSO has an order of magnitude lower error compared to RTTOV13.

Figure 3 shows the average absolute BT error between the Fast-RT model and the LBLRTM model, while Fig. 4 as the maximum absolute error across all profiles and viewing angles. It can be observed that the average brightness temperature shows a degradation in the proposed methods compared to RTTOV v13. In the worst case, the degradation is 0.03 K for channel M7, while the improvement/degradation for the other channels remains below 0.005 K for RTTOV13+SI and 0.038 K for RTTOV13+SI+LASSO. Regarding the maximum absolute error per profile and viewing angle, the predictions of the proposed methods for brightness temperature compared to RTTOV v13 worsen by 1.85 K for channel M7 and 0.41 K for channel M8, the worst cases, while there is an improvement of 1.65 K in BT prediction with RTTOV13+SI+LASSO for channel M9. For the remaining channels, the improvement/degradation stays below 0.15 K for RTTOV13+SI and 0.38 K for RTTOV13+SI+LASSO. In relative terms, these variations are not significant, as shown in Table 8.





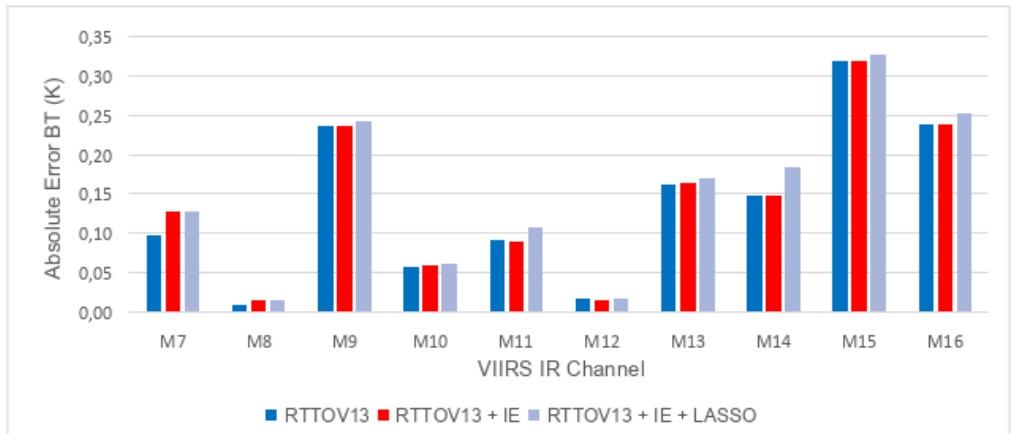

**Figure 3.** Average Absolute Errors in Brightness Temperature (K) between the Fast-RT and LBLRTM models.

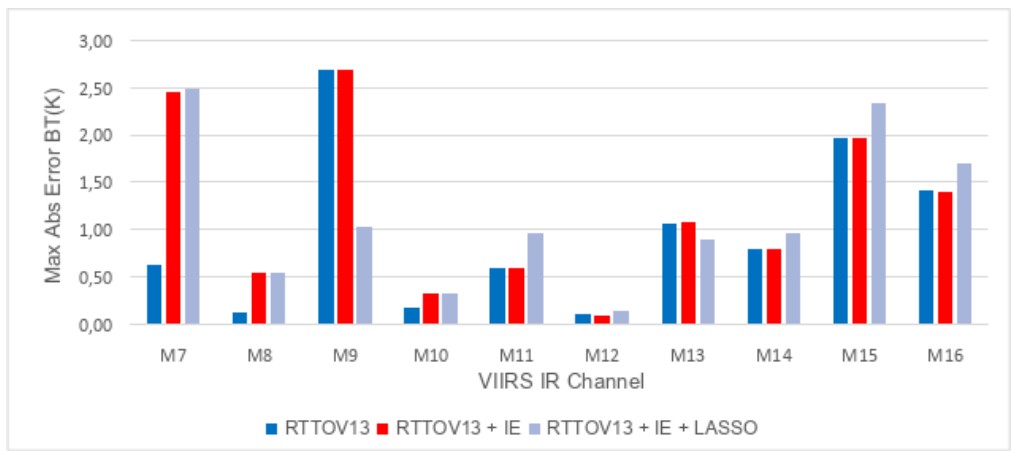

**Figure 4.** Maximum Absolute Errors in Brightness Temperature (K) between the Fast-RT and LBLRTM models.

These findings suggest that while the proposed methods are generally comparable to RTTOV v13 in terms of accuracy, there are specific channels where improvements or further adjustments in the statistical threshold parameters may be necessary to enhance precision if needed.

## 6 Conclusions

This study introduces an automatic and sparse optical depth parametrization method for the RTTOV v13 model to optimize parameter adjustment. The method first applies statistical thresholding across different pressure levels, followed by LASSO regression instead of the traditional least squares approach within the RTTOV v13 framework. This approach enforces significant sparsity across all parameters, leading to a substantial reduction in the computational cost of the Fast-RT model without



significant loss of accuracy, demonstrating strong potential for satellite data assimilation problems. Validation experiments
were conducted on the infrared channels of the VIIRS instrument, with similar results expected for all multispectral infrared
sounders.

The induced sparsity enables the automatic exclusion of gases with negligible absorptivity in a channel, identifies pressure
levels where gases exhibit significant radiance absorption, highlights the most relevant predictors for each gas type, and classi-
fies gases as either fixed or variable. This technique is particularly beneficial for multispectral instruments with channels where
multiple gases strongly correlate with radiance absorption, especially in large-scale variable retrievals for inverse problems.
The proposed method can be extended to other Fast-RT models, such as the Community Radiative Transfer Model (CRTM),
and to other satellite instruments, such as the Advanced Technology Microwave Sounder (ATMS) and the Cross-track Infrared
Sounder (CrIS), to improve the computational efficiency of the radiative transfer model and the accuracy of the retrieved
atmospheric profiles.





## 425 Appendix A: Sparsity Pattern for RTTOV13+SI+LASSO

**Figure A1.** Sparsity pattern for channels M7 to M11 in RTTOV13+SI+LASSO







**Figure A2.** Sparsity pattern for channels M13 to M16 in RTTOV13+SI+LASSO



*Author contributions.* JC conceptualized the study, contributed to the development of the methodology, and supervised the research. FV developed the model code, performed the simulations, and designed and carried out the experiments. JC and FV validated the results, ensuring reproducibility. FV wrote the original draft, while JC was responsible for review and editing.

*Competing interests.* The authors declare that they have no conflict of interest.

*Code and data availability.* The code and data used in this study are available at https://doi.org/10.5281/zenodo.14990818 (Vargas Jiménez and De los Reyes (2025)). Access to the code and data is available to anyone with the provided link, and there are no temporal embargoes or restrictions on access. Anyone who views or downloads the code and data from Zenodo does so anonymously. The software is still under development and is not finalized for end-user applications, but it is provided to allow for the reproduction of the results presented in the
manuscript. The code and data is licensed under the Creative Commons Attribution-NonCommercial 4.0 International (CC BY-NC 4.0) license.

*Acknowledgements.* This research has been supported by Escuela Politécnica Nacional de Ecuador under award PIGR-22-01: *Asimilación de datos satelitales para el sistema de pronóstico meteorológico METEO: selección óptima de predictores y localización óptima de observaciones*. Franklin Vargas acknowledges partial support from the PhD Program in Applied Mathematics, Escuela Politécnica Nacional de
Ecuador. This manuscript benefited from the use of artificial intelligence tools for style correction.



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
