# Peer review of "Automatic Optical Depth Parametrization in Radiative Transfer Model RTTOV v13 via LASSO-Induced Sparsity for Satellite Data Assimilation"

_EGUsphere, 2025_

## Referee Comment (RC1)

Summary

This manuscript presents an approach to reduce the complexity of radiative transfer modeling in RTTOV v13 by combining statistical thresholding with LASSO-induced sparsity for automatic gas selection and predictor reduction. The method is applied to VIIRS channels, with validation against LBLRTM for transmittance and brightness temperatures.

Major Comments

The manuscript presents a technically sound implementation of LASSO-based sparsity within RTTOV v13, but the scientific contribution and practical validation fall short in several areas.

First, while the authors provide general background on radiative transfer models and cite CRTM and previous versions of RTTOV, there is no direct comparison or benchmarking of their approach against existing fast RTMs. This omission limits the reader's ability to evaluate the benefits or drawbacks of the proposed method relative to established techniques.

Second, the method relies on threshold parameters to determine the relevance of gases, yet there is no guidance or sensitivity analysis provided on their selection. Since the method's validity depends on safely discarding certain absorbers, this is a critical omission. Additionally, the manuscript does not discuss scenarios where the assumptions of the method might break down — for example, in unusual atmospheric compositions, extreme pollution events, or volcanic emissions.

Third, LASSO parameter tuning is conducted via a basic grid search, but the authors do not provide any justification for this choice nor discuss why alternative standard methods (such as cross-validation) were not pursued.

Finally, while the authors claim that their approach leads to a "substantial reduction in computational cost," no quantitative analysis is provided to support this. There are no measurements or estimates of runtime or memory savings, nor is there any discussion of what constitutes an acceptable or unacceptable reduction in accuracy for practical applications. The results show mixed performance — with improved RMSE in some VIIRS bands but increased errors in others — but there is no clear guidance on when the method is expected to perform well or poorly.

Structural Comment

The manuscript currently devotes substantial space in Sections 2 and 3 to background material on radiative transfer theory, line-by-line modeling, and general Fast-RT model formulations. While this content is clearly written and technically accurate, much of it summarizes well-established concepts that are not essential for understanding the specific methodological contribution of this paper. The level of detail presented here feels more appropriate for a thesis or tutorial-style document rather than a journal article focused on a specific methodological advance. To improve readability and focus, I recommend substantially condensing these sections in the main text or moving parts of them to an appendix. This would allow the reader to reach the core methodological development (Section 4) more efficiently, without sacrificing completeness for readers who may need additional background.

Minor Comments

L.10: retrieval -> retrievals

L.28: model -> modeling

L.38: add PCRTM: "Liu, X., Smith, W. L., Zhou, D. K., Larar, A. M., Huang, H.-L., Ma, X., & Strow, L. L. (2006). Retrieval of atmospheric profiles and cloud properties from IASI spectra using super-channels. Atmospheric Chemistry and Physics, 6, 255–265. https://doi.org/10.5194/acp-6-255-2006"

L.39: Even though RTTOV is more efficient than line-by-line models, it remains prohibitively expensive for operational  **use cases.**

L.42: RT model.

L.43: model  less computationally

L.44: These decisions must account for the multitude of possible combinations and trade-offs,  **and are typically made by** expert teams .

L.50: cite or remove 'various large-scale applications'

L.51 – 57: mentiones multiple papers that perform 'variable selection' without much context. Not sure what to do with this information.

L.58: specify what those 'parameters' are in Fast RT models.

L.64: Has LASSO been applied to other RTM models?

L.65: Remove section 1.1 title.

L.82: what is 'carbon powder'?

L135: ml is  **the** number

L245: how that value chose? mse(lambda) < **2** mse(0)

L291 – 299: move to appendix

L375: "This suggests that the inclusion of statistical thresholds and LASSO regression in RTTOV v13 slightly affects the accuracy of the transmittance approximation, either improving or worsening it, but the overall variation in error remains negligible." The error in M7 and M8 increases by about 40% with the LASSO method. Why is that negligible?

L395: Don't understand the 'order of magnitude' comparison. For M10 the error increases from 0.89 to 1.4

L406: "These findings suggest that while the proposed methods are generally comparable to RTTOV v13 in terms of accuracy, there are specific channels where improvements or further adjustments in the statistical threshold parameters may be necessary to enhance precision if needed." This work should have been part of this study.

---

## Referee Comment (RC3)

**Review of "Automatic Optical Depth Parametrization in Radiative Transfer Model RTTOV v13 via LASSO-Induced Sparsity for Satellite Data Assimilation" by Franklin Vargas Jiménez and Juan Carlos De los Reyes**

This study applies the LASSO regression to reduce the number of predictors used to calculate optical depth. This makes the RTTOV model more computationally efficient for simulating top-of-atmosphere radiances observed by satellites. The authors demonstrate through experiments that their approach produces good estimates of transmittance and brightness temperatures when compared to the use of the full set of predictors in RTTOV v13. However, these experiments might not be enough to guarantee good performance in the application of satellite data assimilation.

**General comments:**

1. **Application to satellite data assimilation:** The authors suggest that their approach is intended for use in satellite data assimilation. However, they do not present any data assimilation experiments. Such experiments are important, as there may be problems when using this approach in such applications. In data assimilation, the difference between observed satellite radiances and those simulated from model state variables (via RTTOV) is used to update the relevant model state variables. If LASSO induces sparsity by zeroing out many regression parameters, it removes the sensitivity of radiances to certain model variables or layers. As a result, assimilating those radiances may influence fewer aspects of the model state in terms of both variable type and vertical level. This could be undesirable. Therefore, data assimilation experiments are necessary to assess how the induced sparsity affects the assimilation of radiances.

2. **Further explanation and interpretation of results**: In Sections 5.3 and 5.4, the authors provide a numerically detailed discussion of the approximation errors introduced by LASSO. However, it would be beneficial to provide a theoretical interpretation of these results. Specifically, is there a link between the approximation error and the number of non-zero parameters and the characteristics of individual channels? Do the authors believe that the observed variations in performance are largely due to random effects?

3. **Acronyms:** There are lots of acronyms used in the paper. While some might be familiar to many readers, it would be helpful if the authors could provide the full name where they first appear. This applies to both the abstract and the main text.

4. **Formatting issue with citations**: This issue appears in many places (e.g., lines 20-24, lines 32-35 and lines 54-57). For example, on line 54, it should be "… optical images (Hong and Kong, 2021) …"

**Specific comments:**

1. Line 1: This sentence is slightly misleading. In data assimilation, radiative transfer models map model state variables (e.g., temperature) onto the radiances measured by the satellite. It is the radiances that are assimilated, rather than the retrieved temperature.

2. Line 11: Move "(RT)" forward to be after "radiative transfer"

3. Line 11: Again, this sentence is a bit confusing. What the authors describe in the following two paragraphs is exactly what the reviewer expected!

4. Lines 39-40: Even for large centres where RTTOV is being used operationally, the proposed approach has benefits if it reduces computation costs while maintaining accuracy.

5. Line 64: Could the authors provide slightly more clarification at line 50, where it states that LASSO regression has been applied in the context of radiative transfer in Cardall et al. (2023).

6. Section 1.1: The reviewer recommend reformatting this subsection to the last paragraph of Section 1, as there is no Section 1.2.
7. Line 74: Reference for the monochromatic radiative transfer equation (Equation 1).
8. Line 134: Could the authors provide an example of the predictors for a given instrument and gas?
9. Equation (12): The second case is confusing because it states that $d_1 = \bar{d}_1$. How is $d_1$ on the right-hand calculated?
10. Line 232: Readers could benefit from some further discussion on the selection of the thresholds $\epsilon_1$ and $\epsilon_2$.
11. Line 245: Why is a factor of 2 used?
12. Line 261: The full name of VIIRS should be provided earlier in the text.
13. Line 267: What does "SRF" stand for? Does it stand for "Spectral Response Function"?

**Technical corrections:**
1. Line 135: "de number of predictor" -> "the number of predictors"
2. Caption of Table 9: "Maximun Relative Errors" -> "Maximum Relative Errors"
3. Line 172: "… predicted by the model (8)." -> "… predicted by the model (Equation 8)."
4. Line 207: "… considering $M$ angles and $N$ atmospheric profiles …" -> "… considering $N$ angles and $M$ atmospheric profiles …"
5. Line 264: "In this study, we use the VIIRS SRF J2 and can be downloaded from the following link: …" -> "In this study, we use the VIIRS SRF J2, which can be downloaded from the following link: …"

---

## Author Comment (AC1)

**Response to Reviewer 1**

We would like to sincerely thank the reviewer for the thorough evaluation of our manuscript and for the constructive comments and suggestions provided. We deeply appreciate the time and effort devoted to reviewing our work. The feedback has been invaluable and has contributed significantly to improving the quality and clarity of the manuscript. Below, we address each of the comments in detail.

**Major Comments**

- **Reviewer Comment:** *First, while the authors provide general background on radiative transfer models and cite CRTM and previous versions of RTTOV, there is no direct comparison or benchmarking of their approach against existing fast RTMs. This omission limits the reader's ability to evaluate the benefits or drawbacks of the proposed method relative to established techniques.*

  **Authors Response:** The referee is right in noting that a benchmark comparison between established fast RT models—such as CRTM and RTTOV with LASSO-induced sparsity—is of interest. However, this falls outside the intended scope of the present study. Such a comparison would require a dedicated and thorough analysis, given the fundamental differences in their core parameterization strategies: RTTOV employs an additive gas-by-gas optical depth parameterization, whereas CRTM relies on a joint, global parameterization of gas absorptions.

  The main focus of this work is a methodological proposal to evaluate and improve the computational efficiency of the optical depth parameterization in the RTTOV v13 model. This is achieved by replacing the standard OLS regression with LASSO regression to induce sparsity, using inferential statistical techniques to discard gases that are not relevant for the numerical approximation of transmittances, and assessing the advantages and limitations of this approach within the RTTOV scheme.

  Nevertheless, we acknowledge the value of such a comparative study and plan to extend our approach in the future to develop sparsity-driven parameterizations for joint gas absorption schemes, which may be applicable to CRTM or similar frameworks. This represents a promising direction for future research.

- **Reviewer Comment:** *Second, the method relies on threshold parameters to determine the relevance of gases, yet there is no guidance or sensitivity*

*analysis provided on their selection. Since the method's validity depends on safely discarding certain absorbers, this is a critical omission..*

**Authors Response:** We agree that performing a sensitivity analysis on the inclusion or exclusion of individual gases can provide additional insight into their relevance, as has been done in other studies that assess gas importance based on brightness temperature variability.

Currently, in our approach, the decision to discard a gas is not based on arbitrarily chosen statistical thresholds. Instead, it is grounded in the structure of RTTOV's optical depth parameterization and is implemented automatically through a statistical framework based on confidence intervals, as detailed in Subsection 4.1.

Specifically, we compare the results of RTTOV using all gases that exhibit absorption lines in a given channel against those obtained using only the subset selected via our method. In our statistical thresholding approach, we construct confidence intervals for the transmittance at each pressure level for each gas, aiming to contain the true transmittance value with a high confidence level of $p = 1 - \alpha = 1 - 10^{-6}$. This is a deliberately strict threshold, ensuring reliable inferences.

A gas is considered negligible at a given pressure level only if two conditions are simultaneously satisfied: (1) the confidence interval length (or standard deviation of transmittance) is smaller than a relative tolerance $\epsilon_1 = 10^{-6}$, which is stringent considering transmittance lies within the range $(0, 1]$; and (2) the corresponding optical depth is smaller than $\epsilon_2 = 10^{-6}$, allowing us to approximate the transmittance by 1 with a relative error less than $\epsilon_2$, as explained in Case III (2.2.5). Both $\epsilon_1$ and $\epsilon_2$ are fixed relative tolerances, and any future adjustment would likely involve even stricter values.

For a gas to be entirely discarded, both conditions must be satisfied across all pressure levels, meaning that the gas must exhibit transmittance values close to 1 throughout the atmospheric column with a confidence probability of $(1 - \alpha)^{100} \approx 0.9999$ (assuming independence between layers, consistent with the RTTOV parameterization scheme). This constitutes a qualitative assessment informed by rigorous statistical inference; it is not a direct exclusion based solely on the numerical values of threshold parameters. That said, the practical guideline for choosing these parameters, as inferred from their construction and intended purpose, is that they should be sufficiently small.

In the revised version of the manuscript, we extend our numerical results using smaller values for these tolerances, which allow for better fits, and we provide the corresponding analysis.

**Reviewer Comment:** *Additionally, the manuscript does not discuss scenarios where the assumptions of the method might break down — for example, in unusual atmospheric compositions, extreme pollution events, or volcanic emissions.*

**Authors Response:** This is an important and insightful point for the development of a robust Fast-RT framework. The proposed approach could indeed be extended to

handle more complex scenarios, such as treating SO2 as a variable gas as in RTTOV for volcanic environments (rather than a fixed gas as in our current setting), incorporating additional aerosol types to account for extreme pollution events, or refining the treatment of the water vapor continuum, which is not included in our current configuration. Nonetheless, the core idea of our methodology—based on a separation between six variable gases and 22 fixed gases identified as the dominant absorbers—remains valid as the foundation for the proposed absorption parameterization scheme. These potential extensions represent promising directions for future development within the same methodological framework.

However, as stated in our response to the first comment, the present work is primarily a methodological proposal aimed at improving the computational performance of RT-TOV by introducing a sparse regression-based optical depth parameterization for gas absorption and enabling automatic gas selection. The study is not intended to enhance the classical RTTOV performance under diverse or extreme atmospheric conditions. For such considerations, we refer to the official RTTOV v13 Science and Validation Report by Saunders (2020), which defines the core assumptions and validation framework of the model.

- **Reviewer Comment:** *Third, LASSO parameter tuning is conducted via a basic grid search, but the authors do not provide any justification for this choice nor discuss why alternative standard methods (such as cross-validation) were not pursued.*

  **Authors Response:** We appreciate the reviewer's insightful comment regarding our use of grid search to tune the LASSO regularization parameter. Our initial approach aimed to provide a baseline method that is straightforward and widely understood. We acknowledge that alternative techniques, such as cross-validation, are commonly used for parameter tuning; however, in our specific context, cross-validation is not directly applicable. A brief discussion of this point has been added starting at line 243 in the revised manuscript.

  To address this limitation, we will incorporate in a revised version of the manuscript alternative advanced methods, including model selection criteria such as the Bayesian Information Criterion (BIC) and parametric bilevel optimization frameworks. One of the authors has worked extensively on these approaches in recent years, and they offer promising avenues for efficiently and reliably selecting the optimal regularization parameters.

- **Reviewer Comment:** *Finally, while the authors claim that their approach leads to a "substantial reduction in computational cost," no quantitative analysis is provided to support this. There are no measurements or estimates of runtime or memory savings, nor is there any discussion of what constitutes an acceptable or unacceptable reduction in accuracy for practical applications. The results show mixed performance — with improved RMSE in some VIIRS bands but increased errors in others — but there is no clear guidance on when the method is expected to perform well or*

*poorly.*

**Author Response:** The qualitative support for this claim is grounded in the observed sparsity level achieved in the transmittance parameterization. Specifically, the reduction in runtime for evaluating the parameterized transmittance function is directly proportional to the reduction in the number of active parameters, as fewer predictor evaluations are required. Similarly, memory usage is also reduced proportionally; however, in this context, memory savings are of limited practical relevance due to the inherently low memory requirements of a full parameterization. In a revised version of the manuscript, we will refine Section 5.2 to improve clarity by incorporating a quantitative comparison between the percentage reduction in the number of parameters and the corresponding decrease in runtime.

Regarding model performance, we agree that the RMSE of transmittance and BT alone are insufficient to determine whether a Fast-RT model performs well or poorly. For this reason, a second level of validation was included in the manuscript, consisting of the computation of relative errors in BT estimation: average between $\mathcal{O}(10^{-5})$ and $\mathcal{O}(10^{-3})$, and maximum relative errors which are between $\mathcal{O}(10^{-2})$ and $\mathcal{O}(10^{-4})$. From this error, M7 performs poorly for all methods due to an unacceptably large error; the rest of the methods perform comparably to standard RTTOV. These errors are obtained by comparing the BTs predicted by the Fast-RT models against those derived from radiances computed using a Line-by-Line model, which serves as the reference or 'ground truth'. This approach aligns with the core objective of Fast-RT methods: approximating the output of LBL models.

In the revised version of the manuscript, the water vapor continuum absorption was disabled in the LBLRTM output in order to clarify the comparison and improve the consistency between simulations, since this absorption component is not explicitly parameterized in the Fast Radiative Transfer (Fast-RT) models under evaluation. This modification enables a more accurate benchmarking of the line-by-line reference against the Fast-RT approximations for a baseline model, whose methodology can be extended to more comprehensive models in future work. Additionally, the analysis includes a comparison between the brightness temperature residuals and the Noise-Equivalent Differential Temperature (NEdT) of the VIIRS sensor. This evaluation framework serves as an effective diagnostic for validating the performance of the Fast-RT schemes prior to their implementation in satellite radiance assimilation workflows.

**Structural Comment**

**Reviewer Comment:** *The manuscript currently devotes substantial space in Sections 2 and 3 to background material on radiative transfer theory, line-by-line modeling, and general Fast-RT model formulations. While this content is clearly written and technically accurate, much of it summarizes well-established concepts that are not essential for understanding the specific methodological contribution of this paper. The level of detail presented here feels more appropriate for a thesis or tutorial-style document rather than a journal article*

*focused on a specific methodological advance. To improve readability and focus, I recommend substantially condensing these sections in the main text or moving parts of them to an appendix. This would allow the reader to reach the core methodological development (Section 4) more efficiently, without sacrificing completeness for readers who may need additional background.*

**Author Response:** We agree with this observation. In the revised manuscript, we will present the relevant theoretical background on radiative transfer and Fast RT models in a more concise manner, in order to improve readability without sacrificing the necessary foundations to understand the methodological development.

**Minor Comments**

- L.10: *retrieval → retrievals*
  **Done.**

- L.28: *model → modeling*
  **Done.**

- L.38: add PCRTM reference:
  Liu, X., Smith, W. L., Zhou, D. K., Larar, A. M., Huang, H.-L., Ma, X., & Strow, L. L. (2006). Retrieval of atmospheric profiles and cloud properties from IASI spectra using super-channels. *Atmospheric Chemistry and Physics*, **6**, 255–265. https://doi.org/10.5194/acp-6-255-2006
  **Done.**

- L.39: Even though RTTOV is more efficient than line-by-line models, it remains prohibitively expensive for operational  use cases.
  **Done.**

- L.42: RT model, .
  **Done.**

- L.43: model  less computationally
  **Done.**

- L.44: These decisions must account for the multitude of possible combinations and trade-offs,  **and are typically made by** expert teams .
  **Done.**

- L.50: cite or remove 'various large-scale applications'
  **Response**: The references in lines 51–57 pertain to large-scale applications of LASSO for variable selection in the context of radiative transfer.

- L.51–57: mentions multiple papers that perform 'variable selection' without much context. Not sure what to do with this information.
  **Response**: Since the paragraph begins with 'In the context of radiative transfer,' the

citations, as mentioned in the previous comment, refer to large-scale applications of LASSO.

- L.58: specify what those 'parameters' are in Fast RT models.
  **Changes:** ...we target the automatic selection of gases and **optical depth predictors**  in Fast RT models by inducing sparsity in the **weight predictors**  using LASSO regression.

- L.64: Has LASSO been applied to other RTM models?
  **Response**: To the best of the authors' knowledge, the use of LASSO regression specifically for modeling optical depth or transmittance within radiative transfer models has not been previously documented in the literature.

- L.65: Remove section 1.1 title.
  **Done.**

- L.82: what is 'carbon powder'?
  **Response**: "Carbon powder" is a fine particulate form of elemental carbon, typically produced by incomplete combustion or pyrolysis, and includes substances like soot and black carbon that contribute to atmospheric aerosols.

- L.135: ml is  **the** number
  **Done.**

- L.245: how was that value chosen? $mse(\lambda) < \mathbf{2}\, mse(0)$
  **Response**: This represents a relative tolerance that specifies how close the mean squared error (MSE) of the LASSO solution should be to that of the ordinary least squares solution. Since $mse(\lambda) > mse(0) > 0$, the condition can be rewritten as $\frac{mse(\lambda)-mse(0)}{mse(0)} < 1$. We then select the largest value of $\lambda$ among the candidates that satisfies this inequality. This was replaced by a model selector based on the Bayesian Information Criterion.

- L.291–299: move to appendix
  **Not done.**

- L.375: "This suggests that the inclusion of statistical thresholds and LASSO regression in RTTOV v13 slightly affects the accuracy of the transmittance approximation, either improving or worsening it, but the overall variation in error remains negligible."
  The error in M7 and M8 increases by about 40% with the LASSO method. Why is that negligible?
  **Response**: We acknowledge the increase in error for M7 and M8; however, the absolute values of the errors remain small. The overall accuracy of the transmittance approximation is not significantly affected, so we consider the variation to be minor in practical terms. To clarify this in the manuscript, we made the following correction in line 376:  **does not significantly impact the quality of the transmittance approximation**.

- L.395: Don't understand the 'order of magnitude' comparison. For M10 the error increases from 0.89 to 1.4

**Response**: We agree with the observation. The error difference in M10 does not represent an order of magnitude. Therefore, this channel has been excluded from the statement in the revised manuscript.

- L.406: "These findings suggest that while the proposed methods are generally comparable to RTTOV v13 in terms of accuracy, there are specific channels where improvements or further adjustments in the statistical threshold parameters may be necessary to enhance precision if needed."

  This work should have been part of this study.

  **Response**: We agree with the observation. We have added the numerical results for different values of $\epsilon_1$ in the revised manuscript.

---

## Author Comment (AC2)

**Response to Reviewer 2**

We would like to sincerely thank Reviewer 2 for their thorough evaluation of our manuscript and for the constructive comments and suggestions provided. We deeply appreciate the time and effort devoted to reviewing our work. The feedback has been invaluable and has contributed significantly to improving the quality and clarity of the manuscript. Below, we address each of the comments in detail.

**Major Comments**

- **Reviewer Comment:** *Application to satellite data assimilation: The authors suggest that their approach is intended for use in satellite data assimilation. However, they do not present any data assimilation experiments. Such experiments are important, as there may be problems when using this approach in such applications. In data assimilation, the difference between observed satellite radiances and those simulated from model state variables (via RTTOV) is used to update the relevant model state variables. If LASSO induces sparsity by zeroing out many regression parameters, it removes the sensitivity of radiances to certain model variables or layers. As a result, assimilating those radiances may influence fewer aspects of the model state in terms of both variable type and vertical level. This could be undesirable. Therefore, data assimilation experiments are necessary to assess how the induced sparsity affects the assimilation of radiances.*

  **Authors Response:** We thank the reviewer for this insightful and important comment, which touches on a key consideration in the practical application of our method. We fully agree that the use of sparsity-inducing models such as ours in the context of satellite data assimilation requires careful evaluation, especially regarding the sensitivity of simulated radiances to the underlying model state variables.

  As correctly pointed out, LASSO-based regularization, by setting many regression coefficients to zero, may reduce the sensitivity of the forward model to certain variables or atmospheric layers. This could, in turn, limit the ability of the data assimilation system to propagate observational information through the model state, both vertically and across variable types. Therefore, conducting assimilation experiments is indeed crucial to assess how this sparsity impacts the effectiveness of radiance assimilation.

To address this concern and strengthen the manuscript, we will incorporate a new section in the revised version where we evaluate the performance of Fast-RT as a forward operator. While a full data assimilation experiment is beyond the scope of the present study, this first evaluation step aims to provide a diagnostic of the model's realism in simulating satellite radiances.

In particular, we will compare the brightness temperatures (BTs) produced by Fast-RT to those obtained from high-fidelity simulations from LBLRTM. The key criterion we propose is that the absolute difference in BT must be lower than the instrument's noise level: the Noise Equivalent Delta Temperature (NEdT) for the thermal emissive bands (M12–M16), and the Noise Equivalent Delta Radiance (NEdR) for the solar reflective bands (M7–M11). It is clear that any radiance below the instrument noise cannot be detected by it, so the assimilation of these satellite data is sufficient as long as the models, whether Fast-RT or line-by-line, are as accurate as what the instrument can measure.

We suggest that the percentage of atmospheric profiles for which this condition is satisfied constitutes a meaningful and practical metric to evaluate the quality of the forward model. A high proportion of profiles with errors below these thresholds indicates that the model error is smaller than the sensor noise and, therefore, that the simulated radiances are sufficiently accurate for use in satellite retrievals and potentially for data assimilation. This criterion provides a quantitative benchmark aligned with the capabilities of the instrument and the intended application.

We acknowledge that this evaluation does not replace the need for actual data assimilation experiments, which will be an essential next step in future work. Nevertheless, we believe that the proposed analysis offers a relevant and informative proxy for assessing Fast-RT's suitability in assimilation contexts and complements the objectives of reducing computational cost in operational or research-oriented inverse problems.

We will clearly state these points in the revised manuscript, along with the corresponding validation with respect to the VIIRS instrument noise level, and include the results of this performance assessment as a foundation for further developments.

- **Reviewer Comment:** *Further explanation and interpretation of results: In Sections 5.3 and 5.4, the authors provide a numerically detailed discussion of the approximation errors introduced by LASSO. However, it would be beneficial to provide a theoretical interpretation of these results. Specifically, is there a link between the approximation error and the number of non-zero parameters and the characteristics of individual channels? Do the authors believe that the observed variations in performance are largely due to random effects?*

  **Authors Response:** We thank the reviewer for this insightful comment, which deepens the understanding of our model's behavior and highlights an important area for further analysis. We agree that providing a theoretical interpretation of the approximation errors induced by LASSO regularization is essential, especially in relation to the sparsity level and the specific characteristics of each spectral channel.

In particular, and in connection with our first reviewer comment on evaluating model errors relative to instrument noise, we will extend the revised manuscript to include a comprehensive theoretical discussion focused on how the inclusion or exclusion of gases and predictors affects the approximation error. This discussion will clarify how the error depends on the number of non-zero regression parameters and the modeling choices made. We emphasize that the observed differences in error between RTTOV and our Fast-RT model are largely attributable to the tuning of the tolerance parameters $\epsilon_1$ and $\epsilon_2$. To demonstrate this, we will include a table showing how reducing these tolerance thresholds results in lower approximation errors, highlighting the trade-off between achieving sparsity and maintaining accuracy.

Furthermore, for channels close to the visible spectrum (solar reflective bands), the larger errors observed with respect to LBLRTM are not primarily caused by the sparsity induced by LASSO, but rather by simplifications in our radiative transfer model—most notably, the omission of the solar radiation component. Addressing this limitation by explicitly incorporating solar radiation effects is part of our planned future work, which we expect will substantially improve the physical realism and accuracy of our model in these spectral regions.

By including this theoretical analysis and clarifying these aspects, we aim to provide a more complete understanding of the error behavior observed in our results and to set the stage for ongoing improvements.

- **Reviewer Comment:** *Acronyms: There are lots of acronyms used in the paper. While some might be familiar to many readers, it would be helpful if the authors could provide the full name where they first appear. This applies to both the abstract and the main text.*

  **Authors Response:** We thank the reviewer for this helpful suggestion. We acknowledge that the excessive use of acronyms can hinder readability, especially for readers who may not be familiar with all terms. In the revised manuscript, we will ensure that all acronyms are fully spelled out with their corresponding definitions at their first appearance. This will improve the clarity and accessibility of the paper.

- **Reviewer Comment:** *Formatting issue with citations: This issue appears in many places (e.g., lines 20-24, lines 32-35 and lines 54-57). For example, on line 54, it should be "... optical images (Hong and Kong, 2021) ...*

  **Authors Response:** We thank the reviewer for identifying these specific instances. We will carefully review the manuscript to correct the mentioned issues and ensure that citations and phrasing are accurate and consistent throughout the text.

**Specific comments:**

1. Line 1: This sentence is slightly misleading. In data assimilation, radiative transfer models map model state variables (e.g., temperature) onto the radiances measured by the satellite. It is the radiances that are assimilated, rather than the retrieved

temperature.

**Response**: The first sentence of the abstract has been revised to: The assimilation of satellite spectral sounder data  **relies on fast and accurate radiative transfer models to simulate satellite radiances from surface and atmospheric state variables.**

2. Line 11: Move "(RT)" forward to be after "radiative transfer"
   **Done.**

3. Line 11: Again, this sentence is a bit confusing. What the authors describe in the following two paragraphs is exactly what the reviewer expected!
   **Changes**: In satellite data assimilation and remote sensing retrieval, as well as their applications in numerical weather prediction (NWP), the radiative transfer  **(RT) equation is the forward model relating atmospheric state variables to satellite-observed top-of-atmosphere (TOA) radiances across different electromagnetic spectrum channels.**

4. Lines 39-40: Even for large centres where RTTOV is being used operationally, the proposed approach has benefits if it reduces computation costs while maintaining accuracy.
   **Response**: We appreciate the reviewer's comment. Although RTTOV is used operationally in large centers, our approach offers computational savings while maintaining accuracy, benefiting both large centers and smaller agencies.

5. Line 64: Could the authors provide slightly more clarification at line 50, where it states that LASSO regression has been applied in the context of radiative transfer in Cardall et al. (2023).
   **Response**: The input data used were not raw radiance values, but rather reflectance products and other variables derived from different Landsat spectral bands, including ratios and transformations relevant for chlorophyll-a and turbidity detection. While the approach does not model radiative transfer explicitly, it leverages empirical relationships between surface reflectance and in-water constituents.
   **Changes**: LASSO regression was applied by Cardall et al. (2023)  **to estimate water quality parameters such as clarity, temperature, and chlorophyll-a, based on correlations with in situ measurements and near-coincident Landsat spectral data, with a focus on model explainability**.

6. Section 1.1: The reviewer recommend reformatting this subsection to the last paragraph of Section 1, as there is no Section 1.2.
   **Done.**

7. Line 74: Reference for the monochromatic radiative transfer equation (Equation 1).
   **Response**: referenced to Weinreb et al. (1981).

8. Line 134: Could the authors provide an example of the predictors for a given instrument

and gas?

**Response**: Appendix 2 has been added to provide the RTTOV v13 predictors for the gases considered in this study.

9. Equation (12): The second case is confusing because it states that $d_1 = \bar{d}_1$. How is $d_1$ on the right-hand calculated?

   **Response**: If this case occurs, i.e., transmittance shows low variability with respect to atmospheric variables but is not close to 1, then $\bar{d}_1 = -\ln(\bar{\tau}_1)$, where $\bar{\tau}_1$ is the mean transmittance computed across all 83 profiles and all 6 view angles at level 1 of the discretized atmospheric model.

10. Line 232: Readers could benefit from some further discussion on the selection of the thresholds $\epsilon_1$ and $\epsilon_2$.

    **Response**: These statistical threshold tolerances should be close to zero. This clarification is included in the manuscript, and the numerical experiments show results for different values of these tolerances. Moreover, a corresponding analysis is carried out to assess the impact of varying these parameters.

11. Line 245: Why is a factor of 2 used?

    **Response**: This represents a relative tolerance that specifies how close the mean squared error (MSE) of the LASSO solution should be to that of the ordinary least squares solution. Since $mse(\lambda) > mse(0) > 0$, the condition can be rewritten as $\frac{mse(\lambda)-mse(0)}{mse(0)} < 1$. We then select the largest value of $\lambda$ among the candidates that satisfies this inequality. In the revised version of the manuscript, the Bayesian Information Criterion is used as a model selection tool for the same purpose.

12. Line 261: The full name of VIIRS should be provided earlier in the text.
    **Done.** The full name of VIIRS has been added in line 62.

13. Line 267: What does "SRF" stand for? Does it stand for "Spectral Response Function"?
    **Change:** Spectral Response Function (SRF). This is clarified in the revised version of the manuscript.

**Minor Comments**

1. Line 135: "de number of predictor" $\rightarrow$ "the number of predictors"
   **Done.**

2. Caption of Table 9: "Maximun Relative Errors" $\rightarrow$ "Maximum Relative Errors"
   **Done.**

3. Line 172: "... predicted by the model (8)." $\rightarrow$ "... predicted by the model (Equation 8)."
   **Done.**

4. Line 207: "... considering $M$ angles and $N$ atmospheric profiles ..." $\rightarrow$ "... considering $N$ angles and $M$ atmospheric profiles ..."
   **Done.**

5. Line 264: "In this study, we use the VIIRS SRF J2 and can be downloaded from the following link: ..." → "In this study, we use the VIIRS SRF J2, which can be downloaded from the following link: ..."
   **Done.**

---

## Author Response (AR1)

**Response to Reviewer 1**

We would like to sincerely thank the reviewer for the thorough evaluation of our manuscript and for the constructive comments and suggestions provided. We deeply appreciate the time and effort devoted to reviewing our work. The feedback has contributed significantly to improving the quality and clarity of the manuscript. Below, we address each of the comments in detail.

**Major Comments**

• Reviewer Comment: First, while the authors provide general background on radiative transfer models and cite CRTM and previous versions of RTTOV, there is no direct comparison or benchmarking of their approach against existing fast RTMs. This omission limits the reader's ability to evaluate the benefits or drawbacks of the proposed method relative to established techniques.

Authors Response: The referee is right in noting that a benchmark comparison between established fast RT models—such as CRTM and RTTOV with LASSO-induced sparsity—is of interest. However, this falls outside the intended scope of the present study. Such a comparison would require a dedicated and thorough analysis, given the fundamental differences in their core parameterization strategies: RTTOV employs an additive gas-by-gas optical depth parameterization, whereas CRTM relies on a joint, global parameterization of gas absorptions.

The main goal of this work is a methodological approach to evaluate and improve the computational efficiency of the optical depth parameterization in the RTTOV v13 model. This is achieved by replacing the standard least-squares regression with a LASSO regression to induce sparsity, using inferential statistical techniques to discard gases that are not relevant for the numerical approximation of transmittances, and assessing the advantages and limitations of this approach within the RTTOV scheme.

Nevertheless, we acknowledge the value of such a comparative study and aim to extend our approach in the future to develop sparsity-driven parameterizations for joint gas absorption schemes, which may be applicable to CRTM or similar frameworks.

• Reviewer Comment: Second, the method relies on threshold parameters to determine the relevance of gases, yet there is no guidance or sensitivity

analysis provided on their selection. Since the method's validity depends on safely discarding certain absorbers, this is a critical omission.

**Authors Response:** We agree that performing a sensitivity analysis on the inclusion or exclusion of individual gases can provide additional insight into their relevance, as has been done in other studies that assess gas importance based on brightness temperature variability.

In our approach, the decision to discard a gas is not based on arbitrarily chosen statistical thresholds, but rather on the structure of RTTOV's optical depth parameterization. This decision is implemented automatically through a statistical framework based on confidence intervals, as detailed in Subsection 4.1.

Specifically, we compare the results of RTTOV using all gases that exhibit absorption lines in a given channel against those obtained using only the subset selected via our method. In our statistical thresholding approach, we construct confidence intervals for the transmittance at each pressure level for each gas, aiming to contain the true transmittance value with a high confidence level of  $p = 1 - \alpha = 1 - 10^{-6}$ . This is a deliberately strict threshold, ensuring reliable inferences.

A gas is considered negligible at a given pressure level only if two conditions are simultaneously satisfied: (1) the confidence interval length (or standard deviation of transmittance) is smaller than a relative tolerance  $\epsilon_1 = 10^{-6}$ , which is stringent considering transmittance lies within the range (0,1]; and (2) the corresponding optical depth is smaller than  $\epsilon_2 = 10^{-6}$ , allowing us to approximate the transmittance by 1 with a relative error less than  $\epsilon_2$ , as explained in Case III (2.2.5). Both  $\epsilon_1$  and  $\epsilon_2$  are fixed relative tolerances, and any future adjustment would likely involve even stricter values.

For a gas to be entirely discarded, both conditions must be satisfied across all pressure levels, meaning that the gas must exhibit transmittance values close to 1 throughout the atmospheric column with a confidence probability of  $(1-\alpha)^{100} \approx 0.9999$  (assuming independence between layers, consistent with the RTTOV parameterization scheme). This constitutes a qualitative assessment informed by rigorous statistical inference, and not a direct exclusion based solely on the numerical values of threshold parameters.

Authors Change: In the revised version of the manuscript, we extend our numerical results using smaller statistical threshold tolerances, from  $10^{-6}$  to  $10^{-9}$ , which allow for better fits to LBLRTM output and results comparable to RTTOV v13, while maintaining a good sparsity level that translates into reduced runtime. We provide the corresponding analysis for these smaller tolerances.

Reviewer Comment: Additionally, the manuscript does not discuss scenarios where the assumptions of the method might break down — for example, in unusual atmospheric compositions, extreme pollution events, or volcanic emissions.

Authors Response: This is an important and insightful point for the development of a robust Fast-RT framework. The proposed approach could indeed be extended to

handle more complex scenarios, such as treating SO2 as a variable gas as in RTTOV for volcanic environments (rather than a fixed gas as in our current setting), incorporating additional aerosol types to account for extreme pollution events, or refining the treatment of the water vapor continuum, which is not included in our current configuration. Nonetheless, the core idea of our methodology—based on a separation between six variable gases and 22 fixed gases identified as the dominant absorbers—remains valid as the foundation for the proposed absorption parameterization scheme. These potential extensions represent promising directions for future development within the same methodological framework.

As stated in our response to the first comment, the present work is primarily a methodological proposal aimed at improving the computational performance of RTTOV by introducing a sparse regression-based optical depth parameterization for gas absorption and enabling automatic gas selection. The study is not intended to enhance the classical RTTOV performance under diverse or extreme atmospheric conditions. For such considerations, we refer to the official RTTOV v13 Science and Validation Report by Saunders (2020), which defines the core assumptions and validation framework of the model.

• Reviewer Comment: Third, LASSO parameter tuning is conducted via a basic grid search, but the authors do not provide any justification for this choice nor discuss why alternative standard methods (such as cross-validation) were not pursued.

Authors Response: We appreciate the reviewer's insightful comment regarding our use of grid search to tune the LASSO regularization parameter. Our initial approach aimed to provide a baseline method that is straightforward and widely understood. We acknowledge that alternative techniques, such as cross-validation, are commonly used for parameter tuning. In our specific context, cross-validation is not directly applicable. A brief discussion of this point has been added in the revised manuscript.

Authors Changes: In the revised version of the manuscript, we replace the grid search strategy with a bilevel optimization one to optimize the regularization parameter in the LASSO regression model. We propose two variants of the bilevel approach: one inspired by model selection criteria such as the Bayesian Information Criterion (BIC), and the other using an  $\ell_0$  regression for the upper level of the bilevel optimization. One of the authors has worked extensively on these approaches in recent years, and they provide promising avenues for efficiently and reliably selecting the optimal regularization parameters. We also provide extensive numerical experiments demonstrating significant improvements in both model fitting and runtime using this approach.

• Reviewer Comment: Finally, while the authors claim that their approach leads to a "substantial reduction in computational cost," no quantitative analysis is provided to support this. There are no measurements or estimates of runtime or memory savings, nor is there any discussion of what constitutes an acceptable or unacceptable reduction in accuracy for practical applications. The results show mixed performance — with improved

RMSE in some VIIRS bands but increased errors in others — but there is no clear guidance on when the method is expected to perform well or poorly.

Author Response: Our claim is grounded in the observed sparsity level achieved in the transmittance parameterization. Specifically, the reduction in runtime for evaluating the parameterized transmittance function is directly proportional to the reduction in the number of active parameters, as fewer predictor evaluations are required. Similarly, memory usage is also reduced proportionally; however, in this context, memory savings are of limited practical relevance due to the inherently low memory requirements of a full parameterization. In the revised version of the manuscript, we refine Section Sparsity pattern in the parametrization of optical depths to improve clarity by incorporating a quantitative comparison between the percentage reduction in the number of parameters and the corresponding decrease in runtime.

Regarding model performance, we agree that the RMSE of transmittance and BT alone are insufficient to determine whether a Fast-RT model performs well or poorly. For this reason, a second level of validation was included in the revised manuscript, consisting of the computation of relative errors in BT estimation: average between  $\mathcal{O}(10^{-5})$  and  $\mathcal{O}(10^{-3})$ , and maximum relative errors which are between  $\mathcal{O}(10^{-2})$  and  $\mathcal{O}(10^{-4})$ . From this error, M7 performs poorly for all methods due to an unacceptably large error; the rest of the methods perform comparably to standard RTTOV. These errors are obtained by comparing the BTs predicted by the Fast-RT models against those derived from radiances computed using a Line-by-Line model, which serves as the reference or 'ground truth'. This approach aligns with the core objective of Fast-RT methods: approximating the output of LBL models.

**Authors Changes:** In the revised version of the manuscript:**

- 1. We include a comparison of the runtime with the sparsity induced by the different proposed methods, showing the percentage reduction in time relative to the RT-TOV v13 scheme as a function of the induced sparsity when varying the statistical threshold tolerance. The corresponding analysis is also provided.
- 2. The water vapor continuum absorption was disabled in the LBLRTM output to clarify the comparison and improve consistency between simulations, since this absorption component is not explicitly parameterized in the Fast-RT models under evaluation. This modification enables a more accurate benchmarking of the line-by-line reference against the Fast-RT approximations for a baseline model.
- 3. Additionally, we included an analysis comparing the brightness temperature residuals with the Noise-Equivalent Differential Temperature (NEdT) of the M-Band VIIRS sensors. This evaluation framework serves as an effective diagnostic for validating the performance of the Fast-RT schemes and provides guidance on the sensitivity of the different models to simulate small radiances, even those that cannot be measured with instruments.

These modifications, along with the extensive experiments and different levels of validation, show results more consistent with those obtained using the standard RTTOV v13, and demonstrate the potential of our proposed methods for simulating radiances with practical applications in satellite data assimilation.

**Structural Comment**

Reviewer Comment: The manuscript currently devotes substantial space in Sections 2 and 3 to background material on radiative transfer theory, line-by-line modeling, and general Fast-RT model formulations. While this content is clearly written and technically accurate, much of it summarizes well-established concepts that are not essential for understanding the specific methodological contribution of this paper. The level of detail presented here feels more appropriate for a thesis or tutorial-style document rather than a journal article focused on a specific methodological advance. To improve readability and focus, I recommend substantially condensing these sections in the main text or moving parts of them to an appendix. This would allow the reader to reach the core methodological development (Section 4) more efficiently, without sacrificing completeness for readers who may need additional background.

**Author Response:** We agree with this observation. In the revised manuscript, we present the relevant theoretical background on radiative transfer and Fast RT models in a more concise manner, in order to improve readability without sacrificing the necessary foundations to understand the methodological development.

**Authors Changes:** Sections 2 and 3 were combined into a single section in a more compact and concise manner.

**Minor Comments**

Done.

- L.10:  $retrieval \rightarrow retrievals$ Done.
- L.28:  $model \rightarrow modeling$ **Done.**
- L.38: add PCRTM reference: Liu, X., Smith, W. L., Zhou, D. K., Larar, A. M., Huang, H.-L., Ma, X., & Strow, L. L. (2006). Retrieval of atmospheric profiles and cloud properties from IASI spectra using super-channels. *Atmospheric Chemistry and Physics*, **6**, 255–265. https://doi.org/10.5194/acp-6-255-2006
- L.39: Even though RTTOV is more efficient than line-by-line models, it remains prohibitively expensive for operational use in small to medium-sized agency use cases.

  Done.
- L.42: RT model, similar to models based on neural networks. Done.

- L.43: model further less computationally **Done.**
- L.44: These decisions must account for the multitude of possible combinations and trade-offs, which is why large meteorological agencies rely on and are typically made by expert teams to identify an optimal configuration of parameters and gases for the Fast RT model.

Done.

- L.50: cite or remove 'various large-scale applications'

  Response: We have added related citations concerning large-scale applications of the LASSO problem.
- L.51–57: mentions multiple papers that perform 'variable selection' without much context. Not sure what to do with this information.

  Response: Since the paragraph begins with 'In the context of radiative transfer' the

**Response**: Since the paragraph begins with 'In the context of radiative transfer,' the citations in lines 51–5 refer to large-scale applications of LASSO problem.

- L.58: specify what those 'parameters' are in Fast RT models.

  Changes: ...we target the automatic selection of gases and optical depth predictors parameters in Fast RT models by inducing sparsity in the weight predictors parameters using LASSO regression.
- L.64: Has LASSO been applied to other RTM models?

  Response: To the best of the authors' knowledge, the use of LASSO regression specifically for modeling optical depth or transmittance within radiative transfer models has not been previously documented in the literature.
- L.65: Remove section 1.1 title. Done.
- L.82: what is 'carbon powder'?

**Response**: "Carbon powder" is a fine particulate form of elemental carbon, typically produced by incomplete combustion or pyrolysis, and includes substances like soot and black carbon that contribute to atmospheric aerosols.

- L.135: ml is de the number Done.
- L.245: how was that value chosen?  $mse(\lambda) < 2 mse(0)$

**Response**: This represents a relative tolerance that specifies how close the mean squared error (MSE) of the LASSO solution should be to that of the ordinary least squares solution. Since  $mse(\lambda) > mse(0) > 0$ , the condition can be rewritten as  $\frac{mse(\lambda)-mse(0)}{mse(0)}

- 12. Line 261: The full name of VIIRS should be provided earlier in the text. **Done.** The full name of VIIRS has been added in line 62.
- 13. Line 267: What does "SRF" stand for? Does it stand for "Spectral Response Function"? **Change:** Spectral Response Function (SRF). This is clarified in the revised version of the manuscript.

**Minor Comments**

- 1. Line 135: "de number of predictor"  $\rightarrow$  "the number of predictors" **Done.**
- 2. Caption of Table 9: "Maximum Relative Errors"  $\rightarrow$  "Maximum Relative Errors" **Done.**
- 3. Line 172: "... predicted by the model (8)."  $\rightarrow$  "... predicted by the model (Equation 8)."

Done.

- 4. Line 207: "... considering M angles and N atmospheric profiles ..."  $\rightarrow$  "... considering N angles and M atmospheric profiles ..." **Done.**
- 5. Line 264: "In this study, we use the VIIRS SRF J2 and can be downloaded from the following link: ..." → "In this study, we use the VIIRS SRF J2, which can be downloaded from the following link: ..."

  Done.

---

## Author Response (AR2)

Juan Carlos De Los Reyes, Ph.D.
Founding Director and Principal Scientist
MODEMAT Research Center in
Mathematical Modeling and Optimization
Quito, Ecuador
E-mail: delosreyes@modemat.org

Dr. Cenlin He Topic Editor Geophysical Model Development

October 13, 2025

**Response to Reviewer 2 and Editor**

Dear Editor,

We would like to sincerely thank the reviewers for their thorough evaluation of our manuscript and for the constructive comments and suggestions provided. We deeply appreciate the time and effort devoted to reviewing our work. The feedback has contributed significantly to improving the quality and clarity of the manuscript. Below, we address each of the comments in detail.

**Reviewer 2, minor corrections:**

• Reviewer Comment: Because the current title explicitly mentions "data assimilation", readers may expect DA experiments. The authors may consider removing it from the title.

**Response:** We considered the referee's suggestion and removed "for Satellite Data Assimilation" from the title. The new title is: "Automatic Optical Depth Parametrization in Radiative Transfer Model RTTOV v13 via LASSO-Induced Sparsity"

• Reviewer Comment: In the Conclusion section, it would be helpful to add a paragraph that: discusses the potential application of the new approach in DA (the authors' response to the relevant comment could be used here); and clarifies that DA experiments may be conducted in the future.

**Response:** We agree with this suggestion. At lines 528-533, we added the following paragraph:

"The numerical results obtained at different levels of validation, particularly the output from the proposed model indicating a high proportion of profiles with errors below the instrumentâs NEdT, provide strong evidence of its suitability and potential for satellite data assimilation. Nevertheless, applying sparsity-inducing models in this context requires a careful evaluation of the sensitivity of simulated

radiances to the underlying model state variables. This evaluation, in practical scenarios such as the satellite data assimilation of radiances from the proposed Fast-RT model, will be carried out in future work."

• Reviewer Comment: I couldn't verify the code stored in Zenodo (https://doi.org/10.5281/zenodo.14990818), as it is restricted to users with access.

**Response:** The code and data files are already publicly available. The version used for the revised manuscript is available at https://doi.org/10.5281/zenodo.17050361 (September 3, 2025, Version v3). The link to Zenodo has been updated in the manuscript reference.

• Reviewer Comment: In Appendix B, the numbering of the predictors starts at 1, whereas in the figures it starts at 0.

**Response:** The predictor numbered 0 corresponds to the Case II of the statistical threshold parametrization, that is,  $X_{0i} = 1$ , with parameter  $w_{i0}^{\mathbf{g}l} = \overline{d}_i^{\mathbf{g}_l}$ . This is not an original RTTOV v13 predictor, but it has been included in the tables in Appendix B for clarity. The predictor  $\hat{a}0\hat{a}$  has been added to the table with the following caption:

"(\*) Not an original RTTOV v13 predictor. This predictor corresponds to Case II of the statistical threshold parametrization."

**Editor's corrections:**

• Comment: (1) Please address the remaining minor suggestions from Reviewer #2 for the technical corrections.

**Authors Response:** Each of the points suggested by Reviewer #2 has been addressed.

• Comment: (2) Please ensure your discussion on the limitation/future direction of this study mentioned in your responses to Reviewer #1 are also summarized and discussed in your main text, including "no direct comparison or benchmarking of their approach against existing fast RTMs" and "does not discuss scenarios where the assumptions of the method might break down (for extreme conditions)".

**Changes:** We included, in the header of the Numerical Results section (lines 284-286), the following paragraph:

"The numerical experiments do not include direct benchmarking against other existing Fast-RT models, only against standard RTTOV v13. It also does not evaluate scenarios where the assumptions of the method might break down, such as extreme atmospheric conditions, including extreme pollution events and environments with high volcanic activity"

And, in the Conclusion section, as future work (lines 534-536), the following paragraph has been added:

"Additionally, future directions may include a benchmark comparison against other existing Fast-RT models and more general scenarios with extreme atmospheric conditions, considering strong absorption due to extreme pollution events and incorporating variable SO2 concentrations in environments with high volcanic activity."

With kind regards,

Juan Carlos De los Reyes and Franklin Vargas